# Mcm2 promotes stem cell differentiation via its ability to bind H3-H4

Xiaowei Xu[1,2,3,4†], Xu Hua[1,2,3,4†], Kyle Brown[5], Xiaojun Ren[5], Zhiguo Zhang[1,2,3,4]*

[1]Institute for Cancer Genetics, Columbia University Medical Center, New York, United States; [2]Herbert Irving Comprehensive Cancer Center, Columbia University Medical Center, New York, United States; [3]Department of Pediatrics, Columbia University Medical Center, New York, United States; [4]Department of Genetics and Development, Columbia University Medical Center, New York, United States; [5]Department of Chemistry, University of Colorado Denver, Denver, United States

*For correspondence:
zz2401@cumc.columbia.edu

[†]These authors contributed equally to this work

**Abstract** Mcm2, a subunit of the minichromosome maintenance proteins 2–7 (Mcm2-7) helicase best known for its role in DNA replication, contains a histone binding motif that facilitates the transfer of parental histones following DNA replication. Here, we show that Mcm2 is important for the differentiation of mouse embryonic stem (ES) cells. The Mcm2-2A mutation defective in histone binding shows defects in silencing of pluripotent genes and the induction of lineage-specific genes. The defects in the induction of lineage-specific genes in the mutant cells are likely, at least in part, due to reduced binding to Asf1a, a histone chaperone that binds Mcm2 and is important for nucleosome disassembly at bivalent chromatin domains containing repressive H3K27me3 and active H3K4me3 modifications during differentiation. Mcm2 localizes at transcription starting sites and the binding of Mcm2 at gene promoters is disrupted in both Mcm2-2A ES cells and neural precursor cells (NPCs). Reduced Mcm2 binding at bivalent chromatin domains in Mcm2-2A ES cells correlates with decreased chromatin accessibility at corresponding sites in NPCs. Together, our studies reveal a novel function of Mcm2 in ES cell differentiation, likely through manipulating chromatin landscapes at bivalent chromatin domains.

## Editor's evaluation

This manuscript reports a novel role of Mcm2 licensing factor and helicase subunit of the Mcm2-Mcm7 complex in the differentiation of embryonic stem cells into neuronal lineages. A series of compelling experimental manipulations dissect the abnormalities in the formation of heterochromatin at pluripotent genes and the resolution of bivalent chromatin domains at lineage-specific genes in differentiation in response to mutation of the histone binding domain of Mcm2. These findings provide new insights into the replication-independent roles of Mcm2, and will be of interest to scientists working on development and embryonal cell differentiation.

## Introduction

Embryonic stem (ES) cells (ESCs) are pluripotent cells that possess both the ability to self-renew and the potential to differentiate into lineage-specific cell types (*De Los Angeles et al., 2015*). The ability of ESCs to differentiate into specific lineage cell type both in vivo and in vitro opens exciting opportunities to study the events regulating the earliest stages of lineage specification during development (*Keller, 2005*). During mouse ESC differentiation, pluripotency genes such as *Pou5f1*, *Sox2*, and *Nanog* are silenced, whereas lineage-specific genes are up-regulated (*Sha and Boyer, 2008*). These dynamic changes in gene expression are regulated by transcription factors as well as chromatin factors (*Young,*

2011). For instance, the silencing of pluripotency genes (*Pou5f1*, *Sox2*, and *Nanog*) is associated with both a dramatic reduction of tri-methylation of histone H3 lysine 4 (H3K4me3), a histone modification associated with active gene transcription, and an increase in H3K27me3, a repressive histone modification (*Atlasi and Stunnenberg, 2017*; *Mikkelsen et al., 2007*). In contrast, the promoters of lineage-specific genes, which are marked by both repressive H3K27me3 and active H3K4me3 marks, needs to be resolved for silencing and/or induction of lineage-specific genes during differentiation (*Bernstein et al., 2006*; *Harikumar and Meshorer, 2015*). Although significant processes have been made to understand these dynamic changes in chromatin states during differentiation, the regulation of these dynamic changes remains underexplored.

Histone chaperones, a group of proteins best known for their roles in the assembly of DNA into nucleosomes following DNA replication and gene transcription, have been found to play multiple roles in stem cell maintenance and differentiation. For instance, chromatin assembly factor 1 (CAF-1), a histone chaperone involved in the deposition of newly synthesized H3-H4 onto replicating DNA, is important for cell fate maintenance (*Ishiuchi et al., 2015*; *Smith and Stillman, 1989*). Depletion of subunits of the CAF-1 complex in embryonic fibroblasts results in increased reprogramming efficiency into iPSC cells (*Cheloufi et al., 2015*). In ESCs, CAF-1 depletion leads to an increase in the percentage of 2C-like cells (2-cell-stage embryos), as well as defects in differentiation (*Cheng et al., 2019*; *Ishiuchi et al., 2015*). The role of CAF-1 in differentiation and the 2C-like state is linked to the function of CAF-1 in DNA replication-coupled nucleosome assembly. In contrast, Asf1a, a histone chaperone involved in the delivery of newly synthesized H3-H4 to CAF-1 in replication-coupled nucleosome assembly, as well as to HIRA, a histone chaperone involved in replication-independent (RI) nucleosome assembly (*De Koning et al., 2007*; *English et al., 2006*; *Tagami et al., 2004*), regulates the disassembly of nucleosomes at bivalent chromatin domains for the induction of lineage-specific genes (*Gao et al., 2018*). Moreover, histone chaperone CAF-1 promotes the formation of H3K27me3-mediated silencing at pluripotent genes (*Cheng et al., 2019*), whereas HIRA facilitates the PRC2 silencing complex at developmental loci during differentiation (*Banaszynski et al., 2013*; *Ray-Gallet et al., 2002*). The H3.3-HIRA pathway also safeguards identities of differentiated cells, indicating its bimodal role in cell fate transition (*Fang et al., 2018*). Therefore, histone chaperones involved in deposition of newly synthesized H3-H4 play multiple roles in ESC differentiation and maintenance.

In addition to histone chaperones involved in the deposition of newly synthesized histones, we and others have also uncovered a group of proteins that function in recycling parental histone H3-H4 following DNA replication. Pole3 and Pole4, two subunits of leading strand DNA polymerase ε, interact with H3-H4 and facilitate the transfer of parental histones to leading strands of DNA replication forks in both yeast and mouse ESCs (*Li et al., 2020*; *Yu et al., 2018*). On the other hand, Mcm2, a subunit of the minichromosome maintenance proteins 2–7 (Mcm2-7) complex that plays an essential role in DNA replication as the replicative helicase (*Tye, 1999*), contains a histone binding domain (HBD) (*Huang et al., 2015*). Mutations at HBDs (Y81A and Y90A) of Mcm2 (Mcm2-2A) in both yeast and mouse ESCs leads to defects in the transfer of parental histones to lagging strands of DNA replication forks (*Gan et al., 2018*; *Li et al., 2020*; *Petryk et al., 2018*). Moreover, Mcm2 interacts with Asf1 that is bridged by H3-H4 proteins (*Groth et al., 2007*). Besides its functions in DNA replication and histone deposition, Mcm2 also interacts with the carboxyl-terminal domain of RNA Pol II in *Xenopus* oocytes, and the Mcm2-7 complex is required for RNA Pol II-mediated transcription at some settings in mammalian cells (*Snyder et al., 2009*; *Yankulov et al., 1999*), indicating a possible role of Mcm2 in gene transcription.

Mouse ESCs with deletion of Pole3 and Pole4 or mutations at the HBD of Mcm2 grow normally and maintain stemness. As parental histones with their histone modifications are the blueprint for the recapitulation of the epigenetic landscape during cell division (*Corpet and Almouzni, 2009*), we investigated the roles of these parental histone chaperone proteins during differentiation of mouse ESCs. We found that Mcm2, relying on its HBD, promotes mouse ESC differentiation. The Mcm2-2A mutant ESCs show defects both in silencing of pluripotent genes and induction of lineage-specific genes during differentiation. The defects in induction of lineage-specific genes are associated with reduced Asf1a binding in Mcm2-2A mutant cells. Mcm2 localizes at transcription starting sites (TSS) and that this localization is dramatically reduced in Mcm2-2A cells, which correlates with reduced chromatin accessibility at bivalent chromatin domains during differentiation. Together, these studies reveal a novel role for Mcm2 and its ability to bind H3-H4 in the differentiation of mouse ESCs.

## Results

### Mcm2, Pole3, and Pole4 are required for the differentiation of mouse ESCs

CAF-1, Asf1a, and HIRA are histone chaperones involved in deposition of newly synthesized H3-H4, whereas Mcm2, Pole3, and Pole4 are involved in parental histone transfer and recycling (*Serra-Cardona and Zhang, 2018*). It is known that CAF-1, Asf1a, and HIRA function in stem cell differentiation through distinct mechanisms (*Banaszynski et al., 2013*; *Cheloufi et al., 2015*; *Cheng et al., 2019*; *Gao et al., 2018*). We therefore asked whether histone chaperones involved in parental histone transfer are also important in this process. To do this, we first monitored the formation of embryoid bodies (EBs), which mimics the formation of three germ layers in vitro, in Pole3 KO, Pole4 KO, Mcm2-2A single and double mutant mouse ESCs (*Li et al., 2020*; *Figure 1—figure supplement 1A*). Briefly, three-dimensional colonies (EBs) were formed in hanging drops without leukemia inhibiting factor (LIF) for 3 days. Then, EBs were cultured in suspension and collected at different times during differentiation for the evaluation of morphology and expression of selected genes involved in stemness and lineage specification (*Figure 1A*).

We observed that all mutants exhibited reduced EB size at day 9 compared to wild type (WT) cells, with Mcm2-2A, Mcm2-2A Pole3 KO, Mcm2-2A Pole4 KO double mutant cells exhibiting dramatic defects based on morphology (*Figure 1B*, *Figure 1—figure supplement 1A*). Similarly, we observed that the silencing of *Pou5f1,* a gene involved in pluripotency, was compromised in all the mutant cells, with the Mcm2-2A Pole3 KO and Mcm2-2A Pole4 KO double mutant cells showing larger effects than either single mutant alone (*Figure 1C*, *Figure 1—figure supplement 1B*). Finally, the transcription of several lineage-specific genes, representing three germ layers, was delayed in all the mutant cells, with the double mutants showing the strongest defects (*Figure 1C*, *Figure 1—figure supplement 1B*). Together, these studies indicate that Mcm2-2A, Pole3 KO, and Pole4 KO mutations, while having little impacts on the stemness of mouse ESCs, have a profound effect on their differentiation.

Depletion of CAF-1 results in an increase in totipotent 2C-like cells, which are marked by reduced expression of *Pou5f1* and increased expression of the endogenous retrovirus MERVL (*Ishiuchi et al., 2015*). We thus assessed whether Mcm2-2A mutants altered cellular plasticity in ESCs by analyzing the expression of Pou5f1 and MERVL-Gag using immunofluorescence. We found that Mcm2-2A mutant and WT cells showed similarly low frequencies of 2C-like cells (*Figure 1D and E*), indicating that ESC plasticity is not altered in Mcm2-2A mutant cells. Together, these results suggest that the ability of Mcm2 to bind histone H3-H4 plays an important role during mouse ESC differentiation.

### Mcm2-2A mutation compromises mouse ESC differentiation into neural lineages

To further study the function of Mcm2 and Pole4 during ESC differentiation, we differentiated WT, Mcm2-2A, and Pole4 KO single and double mutant ESCs into neural lineage cells as previously described (*Gao et al., 2018*). We then compared expression levels of two pluripotency genes (*Pou5f1* and *Nanog*) and two neural lineage-specific genes (*Sox21* and *Pax6*) in WT, Mcm2-2A, Pole4 KO, and Mcm2-2A Pole4 KO double mutant cells. Consistent with the EB formation assays, Mcm2-2A and double mutant (Mcm2-2A and Pole4 KO) cells exhibited compromised silencing of *Pou5f1* and *Nanog* as well as defects in the induction of *Sox21* and *Pax6* expression (*Figure 2A*, *Figure 2—figure supplement 1A*). In contrast, Pole4 KO had no significant effects on the expression of pluripotency genes or neural lineage genes compared with WT cells during neural differentiation (*Figure 2—figure supplement 1A*).

Because Mcm2-2A mutant ESCs showed consistent defects in differentiation in both EB formation and neural differentiation assays, we focused the remaining studies on this mutant. First, we monitored morphology changes and silencing of the EGFP reporter gene driven by the *Pou5f1* distal enhancer during neural differentiation in WT and Mcm2-2A cells (*Hotta et al., 2009*). The silencing of EGFP reporter and the cell morphologies were comparable between WT and Mcm2-2A cells from day 2 to day 6 (*Figure 2B and C*; *Figure 2—figure supplement 1B*). Interestingly, from day 7 to day 9, we observed a dramatic loss of GFP expression in WT cells, whereas ~30% of Mcm2-2A cells preserved GFP signals at day 9 (*Figure 2B and C*; *Figure 2—figure supplement 1B*). Accordingly, we detected higher expression level of Pou5f1 and Nanog proteins in Mcm2-2A cells compared to WT cells after

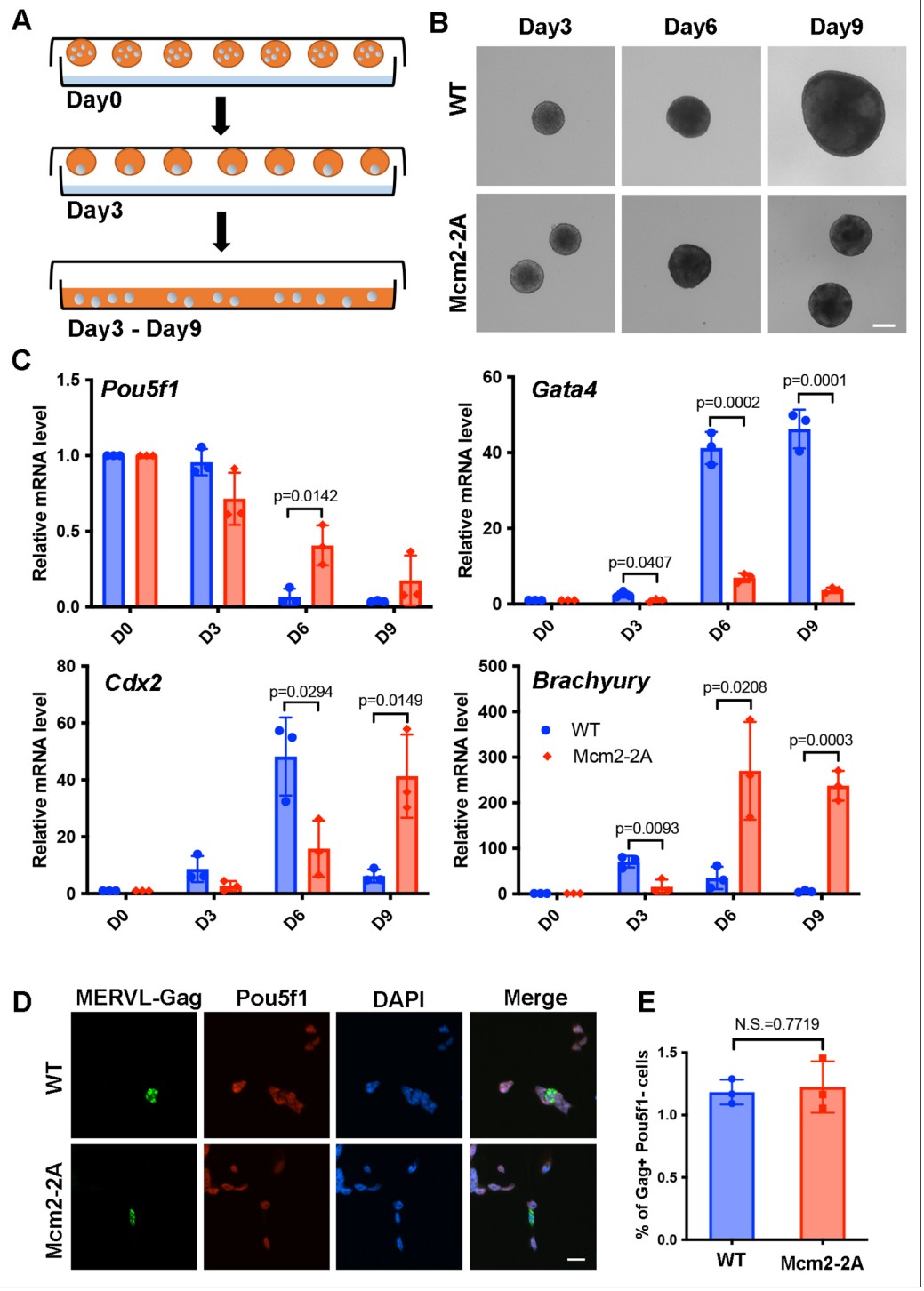

**Figure 1.** Mcm2-2A mutation in mouse embryonic stem cells (ESCs) impairs differentiation. See also *Figure 1—source data 1*. (**A**) A diagram showing the embryoid body (EB) formation assay. (**B**) Representative images of wild type (WT) and Mcm2-2A cells during the process of EB formation. Scale bar: 20 µm. (**C**) RT-PCR analysis of the expression of *Pou5f1* (a gene involved in pluripotency) and three lineage-specific genes in WT and Mcm2-2A cells during EB formation. GAPDH was used for normalization. Data are presented as means ± SD from three independent experiments. (**D**) Representative immunofluorescence images for detection of Pou5f1 and MERVL-Gag proteins in WT and Mcm2-2A mouse ESCs. Scale bar: 20 µm. (**E**) Quantification of Gag+ Pou5f1- cells in (**D**). At least n=1500 cells were counted for each cell line. Data are presented as means ± SD from three

*Figure 1 continued on next page*

*Figure 1 continued*

independent experiments. Statistical analysis in C and E was performed by two-tailed unpaired Student's t test with p values marked on the graphs (N.S., no significant difference).

The online version of this article includes the following source data and figure supplement(s) for figure 1:

**Source data 1.** Relative mRNA level of pluripotency and lineage-specific genes during embryoid body (EB) formation in wild type (WT) and Mcm2-2A cells, and 2-cell (2C)-like ratio in WT and Mcm2-2A embryonic stem cells (ESCs).

**Figure supplement 1.** Mcm2-2A, Pole3 KO, and Pole4 KO mutations affect embryonic stem (ES) cell differentiation based on in vitro embryonic body (EB) formation assays.

**Figure supplement 1—source data 1.** Relative mRNA level of pluripotency and lineage-specific genes during embryoid body (EB) formation in wild type (WT), Pole4 KO, Pole4 KO, Mcm2-2A, Mcm2-2A + Pole3 KO, and Mcm2-2A + Pole4 KO cells.

differentiation (*Figure 2D*). Finally, Mcm2-2A cells upon differentiation did not exhibit neural cell morphology like WT cells (*Figure 2B*), providing additional support to the idea that the Mcm2-2A mutation compromises the differentiation of mouse ESCs.

Next, we tested whether exogenous expression of Mcm2 might rescue defects in differentiation of Mcm2-2A mutant cells (*Figure 2E*). Upon differentiation, we observed that the Mcm2-2A mutant cells expressing WT Mcm2 exhibited typical neural cell morphology similar to that of neural precursor cells (NPCs) generated from WT ESCs (*Figure 2F*). Gene expression analysis by RT-PCR indicated that expression of WT Mcm2 in Mcm2-2A cells led to silencing of *Pou5f1* to a similar degree as WT cells. Interestingly, expression of Mcm2 partially rescued the defects in the expression of lineage-specific genes in Mcm2-2A (*Figure 2G*). These results indicate that defects in differentiation in Mcm2-2A mutant cells can be at least partially rescued by the expression of Mcm2 exogenously.

## Mcm2 and Asf1a function in the same pathway for the induction of lineage-specific genes

Mcm2 interacts with histone chaperone Asf1, which is bridged by H3-H4 (*Huang et al., 2015*). Consequently, the Mcm2-Asf1 interaction is reduced in Mcm2-2A cells. We have shown that deletion of Asf1a, but not Asf1b, in mouse ESCs impairs the induction of lineage-specific genes (*Gao et al., 2018*). To understand how Mcm2-2A affects differentiation, we first overexpressed Asf1a in Mcm2-2A cells and found that Asf1a overexpression could not rescue the defects in differentiation of Mcm2-2A cells based on analysis of cell morphology (*Figure 2—figure supplement 2A, B*) and the expression of *Pou5f1* and lineage-specific genes (*Sox21* and *Pax6*) (*Figure 2—figure supplement 2C–E*). Next, we knocked out Asf1a in either WT cells or Mcm2-2A mutant cells using CRISPR/Cas9 (*Figure 2—figure supplement 2F, G*) and compared the effects of Asf1a KO, Mcm2-2A, and Mcm2-2A Asf1a KO on neural differentiation. We observed that Mcm2-2A Asf1a KO double mutant exhibited similar morphology as Mcm2-2A cells but not Asf1a KO cells after differentiation (*Figure 2—figure supplement 2H*). As reported, Asf1a KO dramatically reduced the induction of lineage-specific genes, while having little effects on the silencing of *Pou5f1* (*Gao et al., 2018*). In contrast, both Mcm2-2A cell line and Mcm2-2A Asf1a KO double mutant cell line showed compromised silencing of *Pou5f1* (*Figure 2—figure supplement 2I*). On the other hand, while significant defects in the induction of neural lineage gene expression (*Pax6* and *Sox21*) were detected in either Mcm2-2A cells or Asf1a KO cells, Mcm2-2A Asf1a KO double mutant cells did not show additional defects in the induction of lineage-specific genes compared with Mcm2-2A or Asf1a KO single mutant alone (*Figure 2—figure supplement 2I*), Interestingly, we observed that Mcm2-2A mutant suppressed the defects in the induction of another neural lineage marker, *Zfpm2* in Asf1a KO cells (*Figure 2—figure supplement 2I*). These results indicate that Mcm2 likely have two roles during differentiation, silencing of pluripotent genes and induction of lineage-specific genes, the latter of which is mediated, at least in part, through Mcm2-H3-H4-Asf1a interaction. This idea was further supported by transcriptome analysis described below.

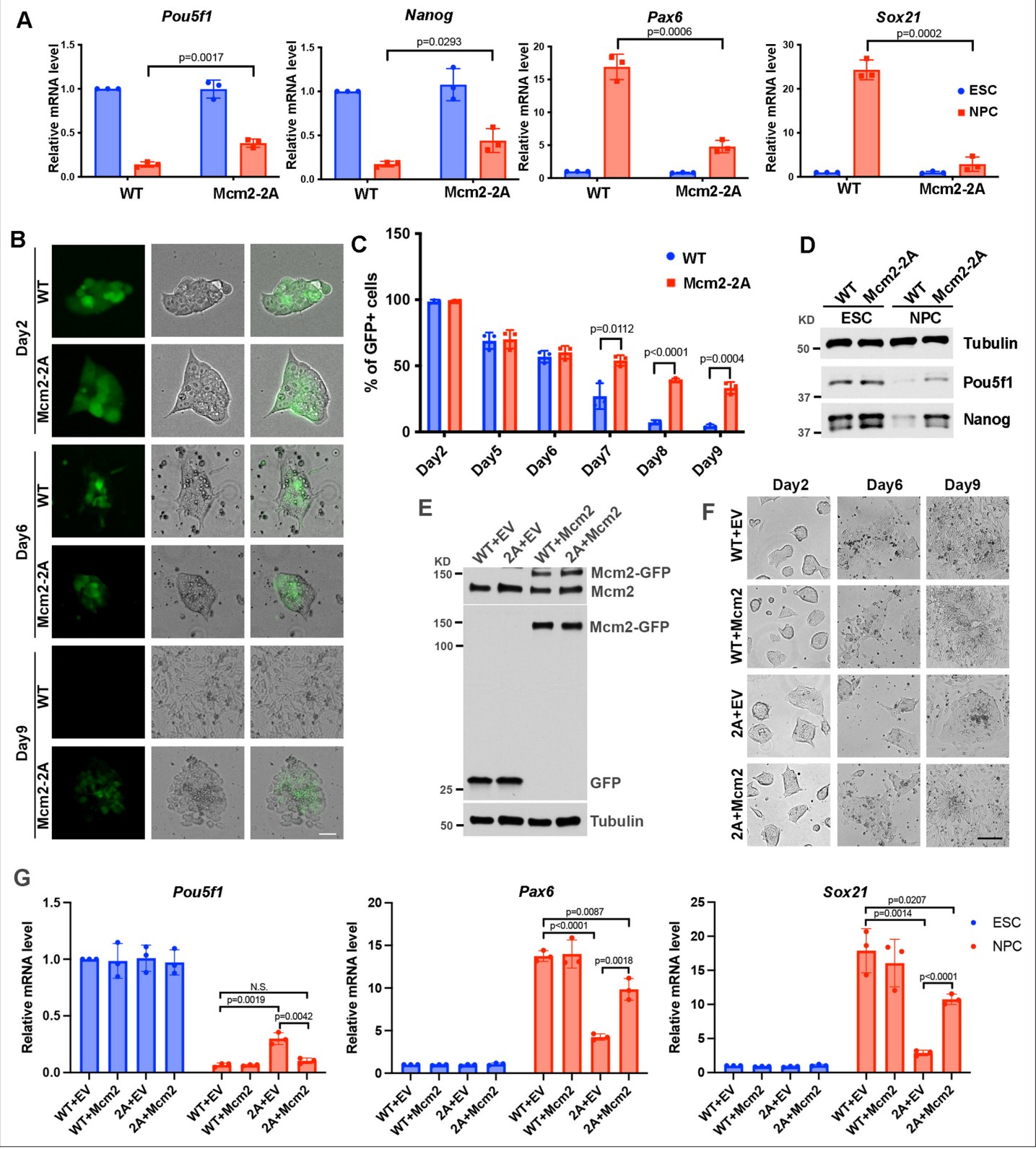

**Figure 2.** Mcm2 is required for neural differentiation of mouse embryonic stem cells (ESCs). See also *Figure 2—source data 1*, *Figure 2—source data 2*, *Figure 2—source data 3* and *Figure 2—source data 4*. (**A**) RT-PCR analysis of expression of two genes involved in pluripotency (*Pou5f1*, *Nanog*) and two neural lineage-specific genes (*Pax6*, *Sox21*) in wild type (WT) and Mcm2-2A cells during neural differentiation. GAPDH was used for normalization. Data are presented as means ± SD from three independent experiments. (**B**) Representative images of WT and Mcm2-2A cells during neural

*Figure 2 continued on next page*

*Figure 2 continued*

differentiation. The expression of EGFP is driven by the *Pou5f1* distal enhancer. Scale bar: 20 µm. (**C**) FACS analysis of the percentage of GFP+ cells in WT and Mcm2-2A cells during neural differentiation. Data are presented as means ± SD from three independent experiments. (**D**) WB analysis of Pou5f1 and Nanog in WT and Mcm2-2A cells of both ESCs and neural precursor cells (NPCs). Tubulin was used as a loading control. (**E**) WB analysis of Mcm2-GFP expression in WT and Mcm2-2A ESCs. pWPXL empty vector (EV) was used as the control. Tubulin was used as a loading control. (**F**) Representative images of EV or Mcm2-GFP expressing WT and Mcm2-2A cells during neural differentiation. Scale bar: 100 µm. (**G**) RT-PCR analysis of expression of *Pou5f1*, *Pax6*, and *Sox21* in EV or Mcm2-GFP expressing WT and Mcm2-2A cells during neural differentiation. GAPDH was used for normalization. Data are presented as means ± SD from three independent experiments. Statistical analysis in A, C, and G was performed by two-tailed unpaired Student's t test with p values marked on the graphs.

The online version of this article includes the following source data and figure supplement(s) for figure 2:

**Source data 1.** Relative mRNA level of pluripotency and lineage-specific genes, and *Pou5f1* enhancer driven EGFP positive cells ratio during neural differentiation in wild type (WT) and Mcm2-2A cells.

**Source data 2.** Whole SDS-PAGE images and uncropped blots represented in *Figure 2D*.

**Source data 3.** Whole SDS-PAGE images and uncropped blots represented in *Figure 2E*.

**Source data 4.** Relative mRNA level of pluripotency and lineage-specific genes in Mcm2 rescued wild type (WT) and Mcm2-2A mutant cells.

**Figure supplement 1.** Mcm2-2A mutant prohibits neural differentiation of mouse embryonic stem (ES) cells.

**Figure supplement 1—source data 1.** Relative mRNA level of pluripotency and lineage-specific genes during neural differentiation in wild type (WT), Pole4 KO, and Mcm2-2A + Pole4 KO cells.

**Figure supplement 2.** Mcm2's function in promoting neural differentiation is partially dependent on Asf1a.

**Figure supplement 2—source data 1.** Whole SDS-PAGE images and uncropped blots represented in *Figure 2—figure supplement 2A and G*.

**Figure supplement 2—source data 2.** Relative mRNA level of pluripotency and lineage-specific genes during neural differentiation in Asf1a rescued or Asf1a KO cells.

## Mcm2-2A mutation perturbs transcription globally during differentiation

To understand how the Mcm2-2A mutation affects ESC differentiation, we used RNA sequencing (RNA-seq) and analyzed the transcriptome of WT and Mcm2-2A mouse ESCs as well as NPCs collected on day 9 of neural differentiation. In general, the Mcm2-2A mutation caused greater gene expression changes in NPCs than in ESCs, with 54 differentially expressed genes (DEGs) between WT and Mcm2-2A ESCs and 609 DEGs between WT and Mcm2-2A NPCs (false discovery rate<0.01, |log2 fold change|>1) (*Figure 3A*), providing an explanation for the observation that the Mcm2-2A mutation does not affect stemness or proliferation of mouse ESCs dramatically. Inspection of RNA-seq data confirmed that the silencing of pluripotency gene *Pou5f1* as well as the induction of two neural lineage genes (*Sox21* and *Pax6*) in Mcm2-2A NPCs was compromised compared to WT NPCs (*Figure 3B*). Unsupervised cluster analysis separated the 648 DEGs identified in Mcm2-2A cells into four groups (*Figure 3C*). Gene ontology (GO) analysis of 352 down-regulated genes in Mcm2-2A NPCs were enriched in neurodevelopment, such as forebrain/hindbrain development and neural nucleus development (*Figure 3D*), whereas 238 up-regulated genes compared to WT ESCs were related to stem cell population maintenance (*Figure 3E*). Of note, the differential effects of Mcm2-2A mutant on the expression of genes in NPCs compared to ESCs likely reflect the role of Mcm2 in the differentiation process, but not necessarily due to differential effects of Mcm2 on ESCs and NPCs. Nonetheless, the Mcm2-2A mutation shows defects in both silencing of pluripotent genes and the induction of lineage-specific genes during differentiation, which in turn contributes to the differentiation defects observed in the mutant cells.

To provide additional insight into defects in gene transcription of Mcm2-2A cells during differentiation, we compared the transcriptome changes caused by Mcm2-2A with those by Asf1a KO in ESCs and NPCs based on published datasets. In both ESCs and NPCs, the number of genes affected by Asf1a KO far exceeded those induced by Mcm2-2A mutation (*Figure 3—figure supplement 1A–D*). Furthermore, the genes whose expression affected by Asf1a KO did not overlap with those affected by Mcm2-2A mutations in ESCs (*Figure 3—figure supplement 1A, B*, ), suggesting that Asf1a likely affects gene expression in ESCs independent of Mcm2. Interestingly, we observed that most of the down-regulated genes in Mcm2-2A mutant NPCs (62% of total), which represent those genes showing defects in induction compared to WT NPC, were overlapped with the down-regulated genes in Asf1a KO NPCs (*Figure 3—figure supplement 1D*), and most of these genes were bivalent genes

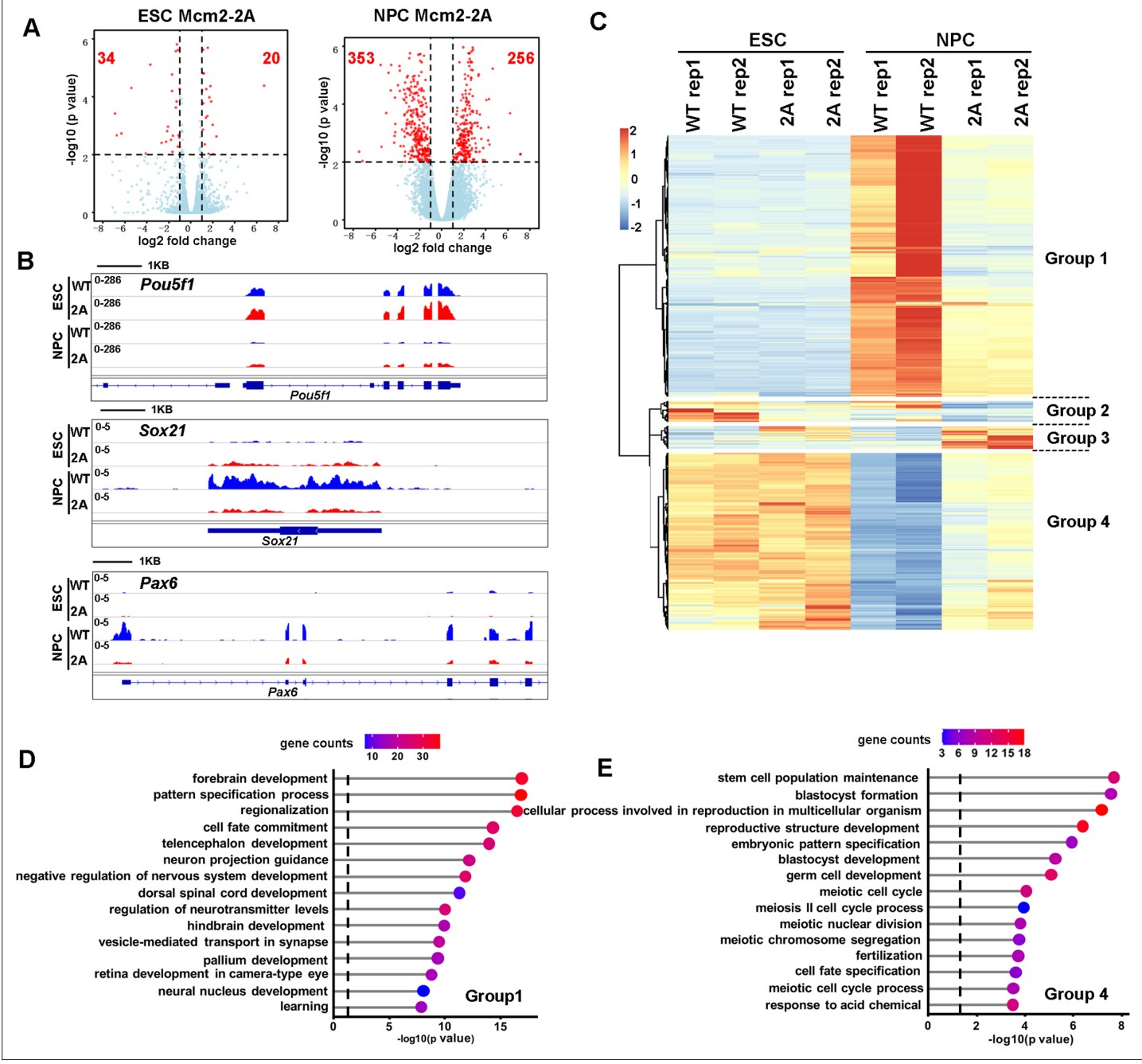

**Figure 3.** Effects of Mcm2-2A mutation on gene transcription in embryonic stem (ES) cells and in neural precursor cells (NPCs). (**A**) Volcano plot of differentially expressed genes (DEGs) between wild type and Mcm2-2A mutant ES cells (left) and NPCs (right) from two independent replicates, with numbers of significantly up-regulated and down-regulated genes (red dots, p<0.01, |log2 fold change|>1) shown. (**B**) RNA sequencing (RNA-seq) tracks showing the sequence read density at *Pou5f1*, *Sox21*, and *Pax6* locus in wild type (WT) and Mcm2-2A ES cells and NPCs. (**C**) The hierarchical clustering analysis of the DEGs between WT and Mcm2-2A cells. RNA expression values (RPKM, reads per kilobase per million reads) are represented by z-score across samples. (**D and E**) Gene ontology (GO) analysis of the group 1 and group 4 genes in (**C**) with the top 15 significant GO terms and p value displayed.

The online version of this article includes the following figure supplement(s) for figure 3:

**Figure supplement 1.** Mcm2 and Asf1a function in the same pathway for the induction of lineage-specific genes.

(*Figure 3—figure supplement 1E*). In contrast, most of the up-regulated genes in Mcm2-2A NPCs compared to WT NPC did not overlapped with those in Asf1a KO cells (*Figure 3—figure supplement 1C*). These results support the idea that Mcm2 and Asf1a function together for the induction of lineage-specific genes expression during differentiation. Collectively, these results suggest that Mcm2's function in promoting ESC differentiation is partially dependent on Asf1a. However, Mcm2 and Asf1a also have independent roles in gene regulation, with Mcm2, but not Asf1a, participating in the silencing of pluripotency genes during differentiation.

## Mcm2-2A mutation disrupts the epigenetic landscape in differentiated cells

During differentiation, histone modification landscapes are rewired (*Mikkelsen et al., 2007*). For instance, bivalent domains that are enriched with both active markers (H3K4me3) and silencing markers (H3K27me3), which are associated with lineage-specific genes, are resolved either through the removal of repressive marker H3K27me3 for gene activation or removal of active marker H3K4me3 for gene silencing (*Voigt et al., 2013*). In addition, pluripotency genes are silenced through gain of H3K27me3 and loss of H3K4me3 at promoters (*Bernstein et al., 2006*; *Harikumar and Meshorer, 2015*). Therefore, we analyzed the impact of the Mcm2-2A mutation on the total levels of H3K27me3 and H3K4me3 during neural differentiation. We observed that in both WT and Mcm2-2A mutant NPCs, the H3K27me3 level was reduced compared to their corresponding ESCs, with a modest reduction of H3K27me3 in Mcm2-2A NPCs compared to WT NPCs (*Figure 4A*). In contrast, the Mcm2-2A mutation did not have an effect on H3K4me3 levels during differentiation (*Figure 4A*). Next, we determined genome-wide distributions of H3K4me3 and H3K27me3 in WT and Mcm2-2A ESCs as well as NPCs using H3K4me3 and H3K27me3 CUT&RUN (*Figure 4B*). Similar to the transcriptome changes, the effects of Mcm2-2A on the chromatin binding of H3K4me3 or H3K27me3 in ESCs were much less pronounced than that in NPCs (*Figure 4B*). Globally, H3K4me3 CUT&RUN density was similar in Mcm2-2A ESCs compared to WT ESCs at TSS of both down-regulated and up-regulated genes in Mcm2-2A ESCs (*Figure 4—figure supplement 1A, B*).

In NPCs, we observed that H3K4me3 CUT&RUN density at the *Pou5f1* gene promoter was dramatically reduced compared to WT ESCs, whereas the level of H3K4me3 at the promoter of *Pax6* was dramatically increased in WT NPCs compared to WT ESCs. These results are consistent with the silencing of *Pou5f1* and induction of *Pax6* during differentiation in WT ESCs. In Mcm2-2A mutant cells, both the reduction of H3K4me3 at the promoter of *Pou5f1* and the increase of this mark at the promoter of *Pax6* were compromised, consistent with the compromised silencing of *Pou5f1* and the induction of *Pax6* in Mcm2-2A mutant cells during differentiation (*Figure 4C*). Furthermore, the average levels of H3K4me3 at the TSS of down-regulated genes in Mcm2-2A mutant NPCs were significantly lower than those of WT NPCs and higher at the TSS of up-regulated genes (*Figure 4D*, *Figure 4—figure supplement 1C*). Together, these results indicate that the gene expression changes in Mcm2-2A mutant NPCs correlate with changes in H3K4me3 levels at gene promoters.

While we observed far more increased H3K27me3 peaks than reduced H3K27me3 peaks in Mcm2-2A mutant NPCs compared to WT NPCs (*Figure 4B*), the changes in gene expression in Mcm2-2A NPCs also correlated with changes of H3K27me3. Specifically, H3K27me3 density at the *Pax6* promoter was higher in Mcm2-2A than WT NPCs, whereas H3K27me3 density at the *Pou5f1* promoter was similar (*Figure 4C*). On average, H3K27me3 density at the TSS of down-regulated genes in Mcm2-2A mutant NPCs was significantly higher than WT NPCs. In contrast, H3K27me3 density at the TSS of up-regulated genes was similar between WT and Mcm2-2A NPCs (*Figure 4E*, *Figure 4—figure supplement 1D*), suggesting that the reduced expression of genes in Mcm2-2A NPCs likely was due to the retention of this repressive mark during differentiation. Collectively, these results indicate that the gene expression alterations in Mcm2-2A mutant NPCs compared to WT NPCs correlate with the changes of both H3K4me3 and H3K27me3 levels at gene promoters.

## Mcm2 is enriched at actively transcribed regions in both ESCs and NPCs

To further explore the mechanisms underlying the transcriptome and epigenetic changes in Mcm2-2A mutant cells, we performed Mcm2 CUT&RUN in both WT and Mcm2-2A ESCs and NPCs. Initial studies of Mcm2 CUT&RUN using antibodies against Mcm2 or against the Flag epitope fused to

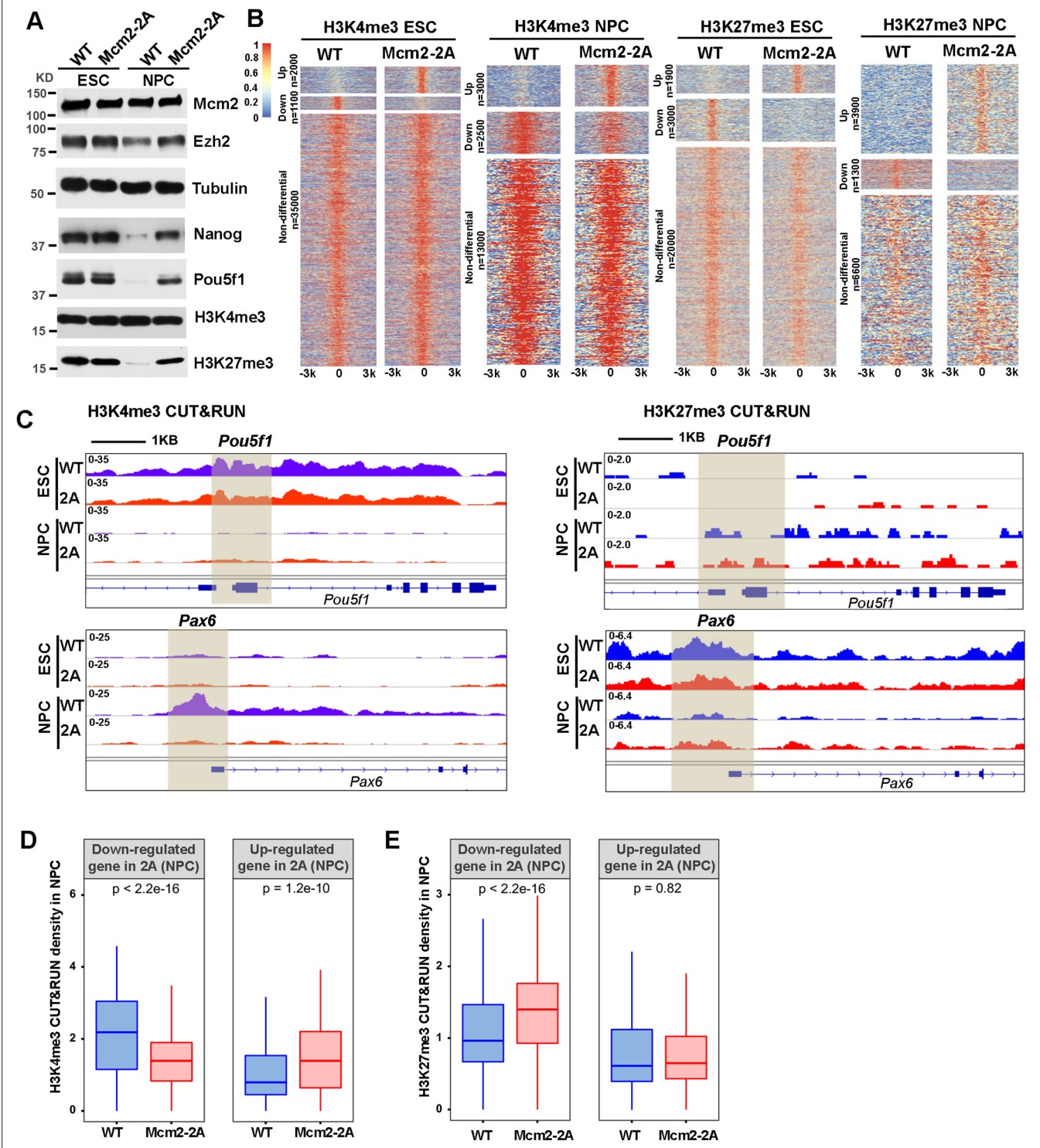

**Figure 4.** Mcm2 mutant affects dynamic changes in H3K4me3 and H3K27me3 during differentiation. See also *Figure 4—source data 1*. (**A**) WB analysis of Mcm2, Ezh2, Nanog, Pou5f1, H3K4me3, and H3K27me3 expression in wild type (WT) and Mcm2-2A cells of both embryonic stem (ES) cells and neural precursor cells (NPCs). Tubulin was used as loading control. (**B**) Heatmaps surrounding CUT&RUN peaks of H3K4me3 and H3K27me3 [–3k, 3k] in WT and Mcm2-2A ES cells and NPCs. n: peak number. Color scale represents reads per million. (**C**) H3K4me3 and H3K27me3 CUT&RUN sequencing density surrounding *Pou5f1* and *Pax6* loci in WT and Mcm2-2A ES cells and NPCs. Shadows indicate CUT&RUN peak signals around the transcription starting site (TSS). (**D and E**) Average of H3K4me3 (**D**) and H3K27me3 (**E**) CUT&RUN density in WT and Mcm2-2A NPCs at the promoters ([–3k, 3k] of TSS) of

*Figure 4 continued on next page*

*Figure 4 continued*

down-regulated and up-regulated genes in Mcm2-2A NPCs (*Figure 3A*, right). The Y-axis represents the log2 ratio of CUT&RUN density (reads per kilobase per million reads [RPKM]). The p values were calculated using Wilcoxon signed-rank test. The average of two independent replicates is shown.

The online version of this article includes the following source data and figure supplement(s) for figure 4:

**Source data 1.** Whole SDS-PAGE images and uncropped blots represented in *Figure 4A*.

**Figure supplement 1.** Effects of Mcm2-2A mutation on H3K4me3 and H3K27me3 distribution during neural differentiation.

---

Mcm2 and Mcm2-2A proteins in ESCs (*Xu et al., 2022*) indicate that Mcm2 antibody and Flag antibody CUT&RUN profiles, each of which had two independent repeats, were highly correlated with each other. Using p=0.0001 as a cutoff, we identified 13742 Mcm2 CUT&RUN peaks in WT ESCs, demonstrating the reproducibility and reliability of Mcm2 CUT&RUN datasets. These peaks likely reflect Mcm2 association with chromatin because control samples (IgG CUT&RUN or Flag CUT&RUN in untagged cells) identified far fewer peaks with the same cutoff, with a large fraction of these peaks did not overlap with Mcm2 CUT&RUN peaks or Flag-Mcm2 CUT&RUN peaks (*Figure 5—figure supplement 1A–C*). We also noticed the relatively low signal-to-noise ratio for the Mcm2 and Flag-Mcm2 CUT&RUN peaks. Once loaded on chromatin during G1 phase of the cell cycle, it is known that the CMG helicase travels along with DNA replication forks (*Prioleau and MacAlpine, 2016*). On the other hand, Mcm2 CUT&RUN experiments were performed using asynchronous ESCs. Therefore, the relatively low signal-to-noise ratio of Mcm2 CUT&RUN peaks likely reflect the dynamic nature of chromatin binding of the CMG helicase during the cell cycle progression. Nonetheless, because of low signal-to-noise ratio, cautions should be made for the interpretation of Mcm2 CUT&RUN results described below.

To gain insight into these Mcm2 CUT&RUN peaks, we aligned and clustered these Mcm2 CUT&RUN peaks based on their overlaps with H3K4me3 and H3K27me3 CUT&RUN signals and ATAC-seq peaks in ESCs (*Figure 5A*). We observed that the majority of Mcm2 CUT&RUN peaks were enriched with H3K4me3 CUT&RUN signals and ATAC-seq peaks. A small number of Mcm2 CUT&RUN peaks were found at bivalent chromatin domains (H3K4me3+ and H3K27me3+), with far fewer peaks co-localizing with H3K27me3 markers (H3K4me3-, H3K27me3+) in ESCs (*Figure 5A*, left). However, when taking into consideration of total number of H3K4me3, H3K27me3, and bivalent domains in ESCs, we observed that over 26% bivalent domains had at least one Mcm2 peaks nearby compared with 21% H3K4me3 peaks having a Mcm2 peak (*Figure 5B*). We also performed similar analysis on 2686 of Mcm2 peaks identified in WT NPCs and observed that almost all the Mcm2 peaks co-localized with H3K4me3 and ATAC-seq peaks (H3K4me3+, H3K27me3-) (*Figure 5A*, right). These results indicate that the chromatin binding of Mcm2 peaks is rewired during differentiation. Moreover, Mcm2 CUT&RUN signals were enriched at the TSS of highly transcribed genes compared to the TSS of lowly transcribed genes in both ESCs and NPCs (*Figure 5C*, *Figure 5—figure supplement 1D*), indicating that the majority of Mcm2 localizes at actively transcribed regions and that Mcm2 may function in gene transcription in a manner independent of its role in DNA replication.

Next, we compared Mcm2 WT and Mcm2-2A mutant CUT&RUN profiles in both ESCs and NPCs. We observed far more reduced Mcm2 CUT&RUN peaks than increased peaks in both Mcm2-2A mutant ESCs and NPCs compared with their WT counterparts (*Figure 5D*; *Figure 5—figure supplement 1C*). Most of these Mcm2 reduced peaks occurred at active promoters and bivalent chromatin regions, whereas increased peaks showed no specific enrichment (*Figure 5—figure supplement 1E*). Because of dramatic changes of Mcm2 localization in both Mcm2-2A ESCs and NPCs, we first asked whether the changes of Mcm2 CUT&RUN peaks in ESCs correlated with those in NPCs. We observed that at the reduced Mcm2 peaks in Mcm2-2A NPCs, Mcm2 CUT&RUN density in Mcm2-2A ESCs was also dramatically decreased compared to WT ESCs, whereas at increased Mcm2-2A CUT&RUN peaks in Mcm2-2A NPCs, Mcm2 density in Mcm2-2A ESCs was higher than in WT ESCs (*Figure 5E*). Together, these results suggest that alterations in chromatin binding of Mcm2-2A mutant proteins at ESCs likely contribute to its altered chromatin binding in NPCs.

We also asked whether the changes in Mcm2 binding in ESCs and NPCs correlated with changes in gene expression in ESCs and in NPCs. We found that the average level of Mcm2-2A density at the TSS of down-regulated genes in ESCs was significantly lower than that in WT ESCs, but Mcm2 density at up-regulated genes was similar between WT and Mcm2-2A mutant ESCs (*Figure 5F*). Similar results

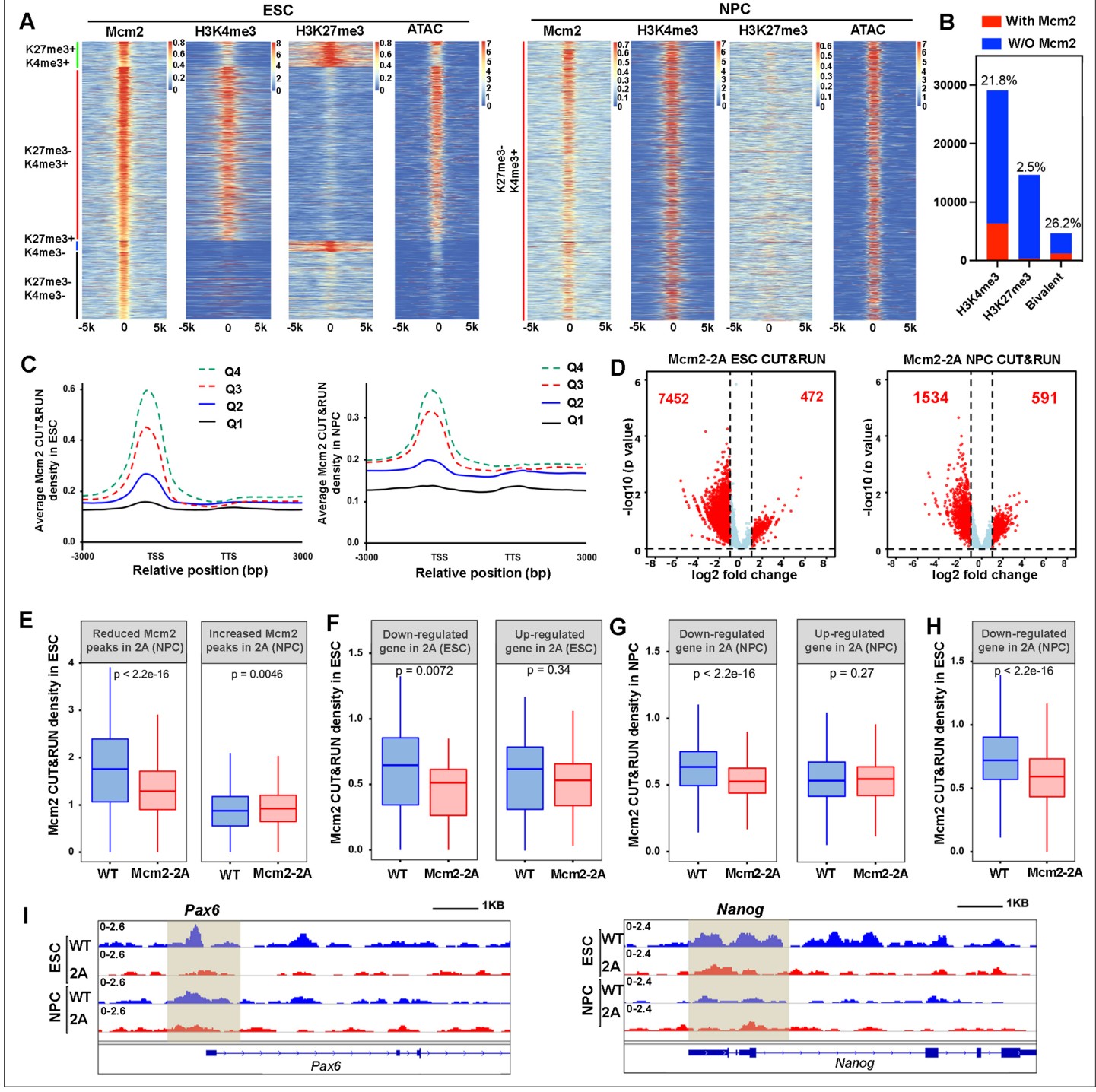

**Figure 5.** Mcm2 chromatin localization largely depends on its ability to bind H3-H4. (**A**) Representative heatmaps of Mcm2, H3K4me3, H3K27me3 CUT&RUN, and ATAC-seq peaks in wild type (WT) embryonic stem (ES) cells (left) and neural precursor cells (NPCs) (right). The density of H3K4me3 and H3K27me3 CUT&RUN and ATAC-seq surrounding Mcm2 CUT&RUN peaks [–5k, 5k] was calculated. Color scales represent reads per million. (**B**) Peak distribution of H3K4me3, H3K27me3, and bivalent domains based on their co-localization with Mcm2 peaks. The percentile of 'with Mcm2 peaks' for each marker were labeled on top. Y-axis represents the peak number. (**C**) Density profiles of Mcm2 CUT&RUN (RPM, reads per million) surrounding transcription starting sites (TSS) and transcription termination sites (TTS). Genes were separated into four groups based on their expression in mouse ES cells or NPCs (Q1=lowest quartile, Q4=highest quartile). (**D**) Volcano plot of differential Mcm2 protein CUT&RUN peaks between WT Mcm2 and Mcm2-2A ES cells (left) and NPCs (right) from two independent replicates, with the total number of significantly up-regulated and down-regulated peaks (|log2 fold change|>1) shown. (**E**) Mcm2 CUT&RUN density in WT and Mcm2-2A ES cells at the reduced and increased Mcm2 CUT&RUN peaks in Mcm2-2A

*Figure 5 continued on next page*

*Figure 5 continued*

NPCs identified in **D**. (**F**) Mcm2 CUT&RUN density in WT and Mcm2-2A ES cells at the promoters ([–3k, 3k] of TSS) of down-regulated and up-regulated genes in Mcm2-2A mutant ESCs (ES cells) identified in *Figure 3A*, left. (**G**) Mcm2 CUT&RUN density in WT and Mcm2-2A NPCs at the promoters ([–3k, 3k] of TSS) of down-regulated and up-regulated genes in Mcm2-2A mutant NPCs (*Figure 3A*, right). (**H**) Mcm2 CUT&RUN density in WT and Mcm2-2A ES cells at the promoters ([–3k, 3k] of TSS) of down-regulated genes in Mcm2-2A mutant NPCs (*Figure 3A*, right). (**E–H**) The Y-axis represents the log2 ratio of CUT&RUN density (reads per kilobase per million reads [RPKM]), with p values calculated using Wilcoxon signed-rank test from two independent replicates. (**I**) Snapshots displaying Mcm2 CUT&RUN density at *Pou5f1* and *Pax6* loci of WT and Mcm2-2A ES cells and NPCs. One representative result from two independent replicates is shown. Shadows indicate CUT&RUN signals around the TSS.

The online version of this article includes the following figure supplement(s) for figure 5:

**Figure supplement 1.** Mcm2-2A mutation decreases Mcm2 binding at chromatin.

were obtained in NPCs (*Figure 5G*). In addition, Mcm2 CUT&RUN density in Mcm2-2A mutant ESCs was reduced at the down-regulated genes in NPCs (*Figure 5H*), as exemplified at the promoter of neural lineage gene *Pax6*, where Mcm2 CUT&RUN density was down-regulated in mutant NPCs compared to WT NPCs (*Figure 5I*). This finding suggests that the reduced association of Mcm2 with chromatin in Mcm2-2A ESCs likely contributes to the reduced expression of these genes during differentiation to NPCs. Taken together, these results indicate that Mcm2 localizes at the TSS of actively transcribed regions in both ESCs and NPCs and that the histone binding of Mcm2 is important for its chromatin binding in ESCs and NPCs. Furthermore, the reduced chromatin binding of Mcm2-2A in ESCs, while having minor effects on gene expression in ESCs, may render these genes susceptible to deregulation during differentiation.

## Mcm2 facilitates chromatin accessibility during mouse ESC differentiation

The landscape of chromatin accessibility dynamically changes during the development (*Trevino et al., 2020*). Given the high correlation of Mcm2 CUT&RUN peaks with ATAC-seq signals in ESCs (*Figure 5A*), we analyzed chromatin accessibility in WT and Mcm2-2A mutant ESCs and NPCs using ATAC-seq. The ATAC-seq repeats in both ESCs or NPCs were highly correlated with each other (*Figure 6—figure supplement 1A, B*). Volcano plot analysis and genome-wide correlation revealed that in ESCs, the Mcm2-2A mutation did not markedly change ATAC-seq profiles (*Figure 6—figure supplement 1A, C*). In contrast, ATAC-seq profiles in Mcm2-2A mutant NPCs were dramatically altered compared to WT NPCs (*Figure 6—figure supplement 1B, D*). These results are consistent with the observations that the Mcm2-2A mutation had little impact on gene expression in ESCs, but markedly altered gene expression in NPCs. Importantly, ATAC-seq signals, which reflect chromatin accessibility, in NPCs were highly correlated with gene expression levels. For instance, we observed an increase and a reduction of ATAC-seq signals at the promoters of *Pou5f1* and *Pax6*, respectively, in Mcm2-2A NPCs (*Figure 6A*). Moreover, the average level of ATAC-seq signals at the TSS of down-regulated genes in Mcm2-2A mutant NPCs was significantly lower than WT NPCs. The opposite was true for up-regulated genes (*Figure 6B*, *Figure 6—figure supplement 2A*). These results are consistent with the idea that chromatin accessibility, as detected by ATAC-seq, is linked to gene transcription.

Next, we explored the correlation between Mcm2 binding and ATAC-seq density. Based on *Figure 5D*, we first separated Mcm2-2A CUT&RUN peaks in ESCs and in NPCs into three categories (increased, reduced, and non-differential Mcm2 peaks compared to WT Mcm2 ESCs and NPCs, respectively) and calculated ATAC-seq density at each group of Mcm2 peaks. We observed a dramatic reduction of ATAC-seq density at the reduced Mcm2 peak group in Mcm2-2A mutant NPCs, along with a slight increase of ATAC-seq density at increased Mcm2 peak groups (*Figure 6C and D*). Of note, ATAC-seq signals at increased Mcm2 peaks were very low (*Figure 6C and D*). In ESCs, ATAC-seq density at down-regulated Mcm2 peaks in Mcm2-2A mutant cells was also reduced, but to a far lesser extent than in NPCs (*Figure 6—figure supplement 2B, C*). These results support the idea that Mcm2 is important for chromatin accessibility in both ESCs and potentially in NPCs.

Since Mcm2 binding in ESCs likely affects gene expression during differentiation (*Figure 5*), we further analyzed the correlation of Mcm2 binding in ESCs with chromatin accessibility during differentiation. Because Mcm2 peaks in ESCs can be classified into four groups based on their co-localization with H3K27me3 and H3K4me3, we analyzed ATAC-seq signals at each group in WT and Mcm2-2A ESCs and NPCs. In ESCs, while Mcm2 CUT&RUN density in Mcm2-2A mutant cells was reduced in all

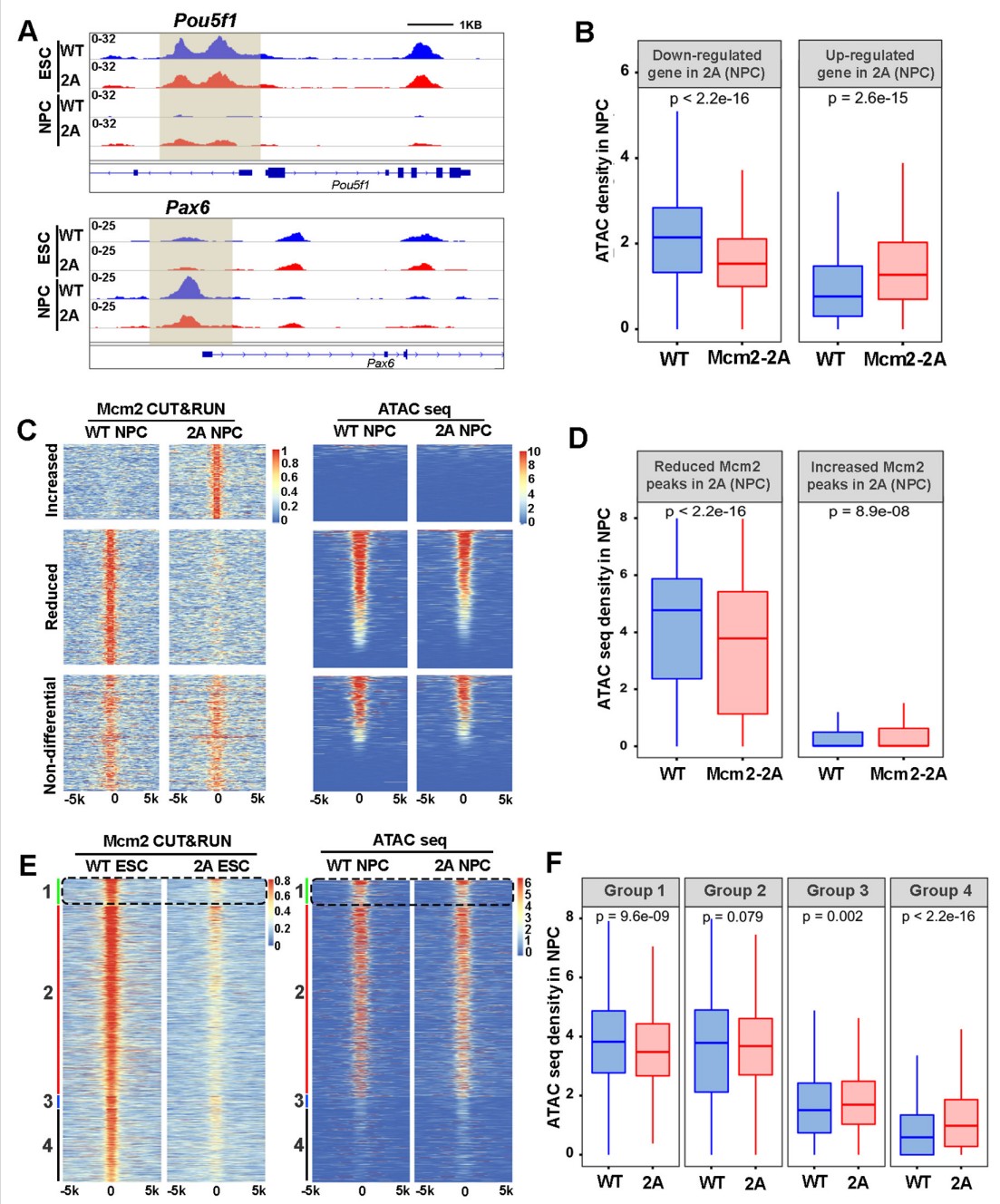

**Figure 6.** Mcm2 facilitates chromatin accessibility during mouse embryonic stem (ES) cell differentiation. (**A**) ATAC-seq tracks displaying ATAC-seq density at the *Pou5f1* and *Pax6* loci in wild type (WT) and Mcm2-2A ES cells and neural precursor cells (NPCs). Shadows indicate CUT&RUN signals around the transcription starting site (TSS). (**B**) ATAC-seq density in wild type (WT) and Mcm2-2A NPCs at the promoters ([–3k, 3k] of TSS) of down-regulated and up-regulated genes in Mcm2-2A NPCs (*Figure 3A*, right). (**C**) Heatmap of ATAC-seq density (right) surrounding Mcm2 CUT&RUN peaks (left, |log2 fold change|>1) in WT and Mcm2-2A NPCs. One representative result from two independent replicates is shown. Color scales represent reads per million. (**D**) The average of ATAC-seq density in WT and Mcm2-2A NPCs at the reduced and increased Mcm2 CUT&RUN peaks in Mcm2-2A NPCs shown in C. (**E**) Representative heatmap of ATAC-seq peaks in WT and Mcm2-2A NPCs at four groups of Mcm2 peaks identified in WT ES cells (*Figure 5A*) based on their co-localization with H3K27me3 and H3K4me3: Group 1: H3K4me3+ and H3K27me3+ (bivalent domain); Group 2: H3K4me3+ and H3K27me3- (active promoters); Group 3: H3K4me3- and H3K27me3+ (repressive promoters); and Group 4: H3K4me3- and H3K27me3-. Color scales represent reads per million. (**F**) Average ATAC-seq density in WT and Mcm2-2A NPCs at each of the four groups shown in E. (**B, D, and F**) The Y-axis represents the log2 ratio of ATAC-seq density (reads per kilobase per million reads [RPKM]). The p values were calculated using Wilcoxon signed-rank test. The average of two independent replicates is shown.

The online version of this article includes the following figure supplement(s) for figure 6:

*Figure 6 continued on next page*

*Figure 6 continued*

**Figure supplement 1.** Chromatin accessibility landscape is altered in Mcm2-2A neural precursor cells (NPCs) compared to wild type (WT) NPCs.

**Figure supplement 2.** Relationships between the impact of Mcm2-2A mutation on Mcm2 binding and chromatin accessibility.

four groups (*Figure 6—figure supplement 2D*), ATAC-seq signals were slightly reduced in Group 2 (H3K4me3+, H3K27me3-) (*Figure 6—figure supplement 2E, F*), suggesting that the reduced Mcm2 binding does not affect chromatin accessibility dramatically. In NPCs, Mcm2 CUT&RUN density in Mcm2-2A mutant cells was reduced in both Group 1 and Group 2, with Group 1 showing a larger reduction than Group 2 (*Figure 6—figure supplement 2G, H*). Importantly, ATAC-seq signals in Mcm2-2A NPCs at Group 1 and Group 2 Mcm2 peaks were significantly reduced compared to WT NPCs, with a larger reduction in Group 1 than in Group 2, whereas ATAC-seq signals in Mcm2-2A NPCs were increased slightly at Group 3 and Group 4 Mcm2 peaks (*Figure 6E and F*). As Mcm2 Group 1 peaks co-localize with bivalent chromatin domains (H3K4me3+ and H3K27me3+), these results indicate that a reduction of Mcm2 binding in Mcm2-2A mutant ESCs, while having little effects on chromatin accessibility in ESCs, perturbs chromatin changes at bivalent chromatin domains during differentiation. Together, these results support the idea that Mcm2 binding at bivalent chromatin domains in ESCs is important for the resolution of these regions for subsequent gene activation during differentiation.

## Discussion

We and others have previously shown that Mcm2, Pole3, and Pole4, three replisome components first known for their roles in DNA replication, function in the transfer of parental H3-H4 following DNA replication in yeast and mouse ESCs (*Gan et al., 2018*; *Li et al., 2020*; *Petryk et al., 2018*; *Xu et al., 2022*; *Yu et al., 2018*). Remarkably, mouse ESCs with deletion of Pole3, Pole4, or with mutations at the histone binding motif of Mcm2 (Mcm2-2A) largely grow normally. Here, we found that these mutant cells all exhibit defects in differentiation, revealing a novel role of these proteins in ES differentiation.

Mcm2 is a subunit of MCM helicase consisting of Mcm2-7. The MCM helicase is loaded on chromatin at the G1/S transition and serves as the core of the CMG replicative helicase to unwind double-stranded DNA for DNA synthesis during the S phase of the cell cycle (*Tye, 1999*). The N-terminus of Mcm2 contains a conserved histone binding motif that interacts with H3-H4 (*Huang et al., 2015*). Mutations at this histone binding motif (Mcm2-2A) that impair Mcm2's ability to bind H3-H4 lead to a dramatic enrichment of parental H3-H4 at leading strands compared to lagging strands of DNA replication forks during early S phase of the cell cycle (*Gan et al., 2018*; *Li et al., 2020*; *Petryk et al., 2018*). Using two different in vitro differentiation assays, we observed that Mcm2-2A mutant mouse ESCs, while growing normally, showed dramatic defects during differentiation. Furthermore, we found that the Mcm2-2A mutation induces global changes in gene expression, chromatin accessibility, and histone modifications during differentiation. Together, these studies reveal a novel role of Mcm2, through its interaction with H3-H4, during differentiation.

Previously, we have shown that deletion of two histone chaperones involved in deposition of newly synthesized H3-H4, Asf1a and the p150 subunit of the CAF-1 complex, also impairs differentiation of mouse ESCs (*Cheng et al., 2019*; *Gao et al., 2018*). Compared to CAF-1 p150 KO and Asf1a KO cells, the Mcm2-2A mutation exhibited distinct defects in differentiation. p150 KO mutant ESCs show defects in both the silencing of pluripotent genes and induction of lineage-specific genes during differentiation, with defects in the former much more pronounced (*Cheng et al., 2019*). In contrast, Asf1a KO ESCs minimally impact the silencing of pluripotent genes and dramatically affect the induction of lineage-specific genes (*Gao et al., 2018*). Silencing of pluripotent genes and induction of lineage-specific genes are both defective in Mcm2-2A cells, with the defects in the latter more pronounced. These results suggest that CAF-1, Mcm2, and Asf1a likely perform non-overlapping functions during differentiation.

First, our results indicate that Mcm2 and Asf1a function in the same pathway for the induction of lineage-specific genes. For instance, we found that Mcm2-2A Asf1a KO double mutant cells showed similar defects in induction of lineage-specific gene as Mcm2-2A and Asf1a KO single mutant alone, suggesting that Mcm2 and Asf1a function in the same pathway for the induction of lineage-specific genes during differentiation. Second, transcriptome analysis by RNA-seq indicate that a large fraction

of genes defective in induction during differentiation were overlapped between Mcm2-2A and Asf1a KO cells. Previously, it has been shown that Asf1 interacts with Mcm2 and this interaction, bridged by H3-H4, is reduced in Mcm2-2A mutant cells (*Huang et al., 2015*). Furthermore, we found that Asf1a is important to resolve bivalent chromatin domains through its ability for nucleosome disassembly (*Gao et al., 2018*). Consistent with this idea, chromatin accessibility at bivalent chromatin domains is reduced the most in Mcm2-2A mutant cells during differentiation, and this reduction is linked to the reduced Mcm2 at bivalent chromatin domains in ESCs, indicating that in addition to its role in parental histone transfer, Mcm2 also has a role in dynamic changes in chromatin during cell fate transition. Therefore, we suggest that Mcm2 and Asf1a function together at bivalent chromatin domains for the induction of lineage-specific genes during differentiation.

Second, our results indicate that Mcm2-2A mutant ESCs also show defects in differentiation in Asf1a-independent manner. For instance, silencing of pluripotent genes such as *Pou5f1* and *Nanog* was defective in Mcm2-2A, but not Asf1a KO cells. Furthermore, Mcm2-2A Asf1a KO double mutant cells also display defects in silencing of *Pou5f1* similar to Mcm2-2A cells. In addition to reduced binding to Asf1a, Mcm2-2A mutant cells are also defective in the transfer of parental H3-H4 to lagging strands of DNA replication forks (*Huang et al., 2015*; *Li et al., 2020*; *Petryk et al., 2018*) in ESCs. Therefore, the defects in silencing of pluripotent genes in Mcm2-2A mutant cells may be due to defects in the recycling of parental H3 during differentiation. Consistent with this idea, mutations at Pole3 and Pole4, two other genes involved in recycling both parental H3.3 and H3.1 in mouse ESCs following DNA replication (*Xu et al., 2022*), also result in defects in differentiation. Currently, the fate of parental histones at different chromatin domains during mouse ESC differentiation is largely unknown. It is possible that the ability of Mcm2 to recycle both parental H3.1 and H3.3 contributes to dynamic changes in chromatin states during differentiation. Alternatively, it is possible that Mcm2, via its ability to bind H3-H4, regulates gene expression directly during differentiation. Supporting this idea, we have shown that Mcm2 localizes at gene promoters of actively transcribed genes in both ESCs and NPCs. Furthermore, MCM2-7 proteins are associated with RNA Pol II in *Xenopus* and HeLa cells (*Snyder et al., 2009*; *Yankulov et al., 1999*), and Mcm2 and Mcm5 are required for Pol II-mediated transcription (*Snyder et al., 2009*). Finally, it is possible that defects in silencing of pluripotent genes in Mcm2-2A mutant cells are associated with firing of dormant origins. It is known that excessive amounts of MCM2-7 complexes than active origins are loaded on chromatin during G1 phase of the cell cycle. Dormant origins provide a first line of defense for the genome under replication stress. ESCs contain more dormant origins than progenitor cells such as NPCs (*Ge et al., 2015*). Consistent with this idea, we detected 13742 and 2686 Mcm2 CUT&RUN peaks, respectively, in WT ESCs and NPCs. Furthermore, activation of dormant origins is impaired in Mcm2 depleted cells (*Ibarra et al., 2008*), and that partial depletion of Mcm4 and Mcm5 does not affect ESCs self-renewal but impairs their differentiation, including toward the neural lineage (*Ge et al., 2015*). However, unlike depletion of MCM subunits in these studies, Mcm2-2A mutation in yeast and mouse ESCs has little impacts on the response to replication stress. Therefore, it is unlikely that defects in differentiation in Mcm2-2A mutant cells are largely due to its impact on firing of dormant origins. Nonetheless, future studies are needed to address whether Mcm2's role in differentiation, in particular its role in silencing of pluripotent genes during differentiation is linked to its ability to bind H3-H4.

In summary, our results indicate that Mcm2, through its interaction with H3-H4, likely regulates differentiation through multiple mechanisms, one of which involves in the resolution of bivalent chromatin domains through its interaction with Asf1a bridged by H3-H4.

## Materials and methods

### Materials availability statement

Further information and requests for resources and reagents should be directed to and will be fulfilled by the Lead Contact, Zhiguo Zhang (zz2401@cumc.columbia.edu).

### Cell culture and cell lines

The mouse E14 ESC line (RRID:CVCL_C320) was kindly provided by Dr Tom Fazzio (University of Massachusetts Medical School) and tested negative for mycoplasma. The stemness of ESCs were analyzed by testing the expression of *Pou5f1*, *Sox2*, and *Nanog*, as well as alkaline phosphatase

staining. In addition, the karyotype of E14 ESCs was also monitored. Cells were grown in DMEM (Corning) medium supplemented with 15% (v/v) fetal bovine serum (GeminiBio), 1% penicillin/strep-tomycin (Gibco), 1 mM sodium pyruvate (Gibco), 2 mM L-glutamine (Gibco), 1% MEM non-essential amino acids (Gibco), 55 µM β-mercaptoethanol (Gibco), and 10 ng/ml mouse LIF (mLIF) on gelatin-coated dishes in the presence of 5% $CO_2$ atmosphere at 37°C.

Mcm2-2A mutant E14 cell lines were generated as described before (*Xu et al., 2022*). To generate Asf1a KO cell lines, sgRNA (5'-GATCACCTTCGAGTGCATCG-3') targeting Asf1a was cloned into the PX459 vector. Targeting plasmids were transfected into mouse ESCs by using Lipofectamine 3000. Clones were identified by Surveyor nuclease assays and the knockout confirmed by Sanger sequencing.

To generate pluripotency EGFP reporter mouse E14 WT and Mcm2-2A ESC line, the lentivirus-based EGFP reporter vector PL-SIN-EOS-C(3+)-EGFP plasmid (21318, Addgene) was used for infec-tion. After selection, single cells were seeded and individual clones were then isolated, expanded, and confirmed under fluorescence microscope.

To generate mouse E14 WT and Mcm2-2A ESC lines expression Mcm2 or Asf1a ectopically, the lentivirus-based vector pWPXL empty vector, pWPXL-Mcm2, or pWPXL-Asf1a plasmids were used for infection. After selection, single cells were seeded and individual clones were then isolated, expanded, and confirmed using Western blot.

## Antibodies

Antibodies used in this study were: anti-MERVL-gag (A-2801, Epigentek), anti-Pou5f1 (sc-5279; Santa Cruz Biotechnology), anti-Nanog (A300-397A, Bethyl), anti-Tubulin (12G10, DSHB), anti-Ezh2 (5246, Cell Signaling), anti-H3K4me3 (ab8580, Abcam), anti-H3K27me3 (9733, Cell Signaling), anti-Flag (F1804, Sigma-Aldrich), anti-Mcm2 (ab4461, Abcam), and anti-Flag (F1804, Sigma).

## EB assay

Mouse ESCs were disaggregated and suspended in ESC medium without LIF. EBs were formed using the hanging drop method (300 cells per drop) on dish lids for 3 days. EBs were then collected and cultured in 10 cm low attachment Petri dish in ESC culture medium without LIF, and the medium was changed every other day. Samples were collected at the indicated time points for analysis of gene expression.

## Immunofluorescence

Cells were seeded on coverslip coated with 1% gelatin, and then fixed in 4% of formaldehyde for 15 min at room temperature (RT). After washing with PBS, fixed cells were permeabilized with 0.1% Triton X-100 in PBS (PBST) for 10 min and blocked for 1 hr with 5% normal goat serum (NGS) in PBST at RT. Cells were incubated with primary antibodies diluted in 1% NGS in PBST overnight at 4°C. Cells were then washed with PBST and incubated with fluorophore-labeled secondary antibodies for 1 hr at RT. DNA were stained with DAPI. Images were captured by Nikon 80i Fluorescence Microscope.

## RT-PCR analysis

Total RNA was isolated from 1×106 cells using RNeasy Plus kit (74104, Qiagen); 0.5 µg of total RNA were used for cDNA synthesis with random hexamers (18080-051, Invitrogen). Quantitative PCR was performed in 12 µL reactions containing 0.1 µM primers and SYBR Green PCR Master Mix (Bio-Rad Laboratories). Primers used are listed below.

| Oligo name | Forward | Reverse |
|---|---|---|
| *Gapdh* RT | CTGACGTGCCGCCTGGAGAAAC | CCCGGCATCGAAGGTGGAAGAGT |
| *Pou5f1* RT | CCCGAAGCCCTCCCTACAGCAGAT | TGGGGGCAGAGGAAAGGATACAGC |
| *Nanog* RT | CCTTCCCTCGCCATCACACT | AGAGGAAGGGCGAGGAGAGG |
| *Cdx2* RT | GCGGCTGGAGCTGGAGAAGGAGTT | CGGCGGCTGTGGAGGCTGTTGT |
| *Brachyury* RT | TCCCGGTGCTGAAGGTAAATGTGT | TTGGGCGAGTCTGGGTGGATGTAG |
| *Nestin* RT | TCGGGAGAGTCGCTTAGAG | AGTTGCTGCCCACCTTCC |

*Continued on next page*

*Continued*

| Oligo name | Forward | Reverse |
| --- | --- | --- |
| *Pax6* RT | TACCAGTGTCTACCAGCCAAT | TGCACGAGTATGAGGAGGTCT |
| *Sox21* RT | GCGGTGCTTTACGATACGTTG | CCGAACATCAGAACCGAGCT |
| *Gata4* RT | CGCCGCCTGTCCGCTTCC | TTGGGCTTCCGTTTTCTGGTTTGA |
| *Zfpm2* RT | TGAAGACAACTCGCATCAGG | TAGCTCCCTCTGGGTCTGAA |

## Neural differentiation assay

The neural differentiation of ESCs was performed as previously described with some modifications (*Gao et al., 2018*; *Ying et al., 2003*). Briefly, ESCs (day 1) were seeded at high density (2×10$^6$ cells/60 mm dish) onto gelatin-coated dish in standard mES medium with LIF for 24 hr. The differentiation was initiated by seeding ESCs (day 2) in N2B27 medium at a density of 3×10$^5$ cells per 60 mm dish. The N2B27 medium was changed every day up to day 6. Cells were cultured continuously in N2B27 medium supplied with EGF (10 ng/ml, R&D) and FGF-2 (10 ng/ml, R&D) for 3 more days with medium changed every day. Neural precursors were collected at day 9 for analysis.

## FACS analysis of GFP ratio

For GFP ratio analysis, exponentially growing mouse ESCs expressing EGFP reporter gene driven by *Pou5f1* distal enhancer were collected and washed in PBS. Samples were analyzed by Attune NxT software of Attune flow cytometer (Thermo Fisher Scientific). Data were analyzed by FCS Express (version 7).

## CUT&RUN

CUT&RUN was performed by following a published protocol with some modifications (*Skene et al., 2018*). Cells were fixed in 0.5% PFA for 2 min and washed three times with washing buffer (20 mM HEPES-NaOH pH 7.5, 150 mM NaCl, 1% Triton X-100, 0.05% SDS, 0.5 mM spermidine and 1× proteinase inhibitor cocktail) and immobilized to concanavalin A-coated magnetic beads. Cells were then incubated overnight at 4°C with primary antibody (1:400 for Mcm2, 1:1000 for Flag, 1:1000 for H3K4me3, and 1:400 for H3K27me3) in antibody binding buffer (20 mM HEPES-NaOH pH = 7.5, 150 mM NaCl, 0.5 mM spermidine, 1% Triton X-100, 0.05% SDS, 2 mM EDTA, 0.04% digitonin, and 1× proteinase inhibitor cocktail). After washing with dig-wash buffer (0.04% digitonin in washing buffer), cells were incubated with pre-assembled second antibody + pA-MNase complex in dig-wash buffer for 1 hr at 4°C. After washing unbound pA-MNase, 2 mM CaCl$_2$ was added to initiate digestion at 0°C for 30 min. Reactions were stopped by mixing with 2× Stop buffer (340 mM NaCl, 20 mM EDTA, 4 mM EGTA, 0.05% digitonin, 25 μL 100 μg/ml RNase A, 1% Triton X-100, and 0.05% SDS) followed by incubation at 37°C for 30 min. DNA in supernatant was mixed with same volume of 2× elution buffer (20 mM Tris-HCl pH = 8.0, 20 mM EDTA, 300 mM NaCl, 10 mM DTT, and 2% SDS) and reverse cross-linked at 65°C overnight. DNA was purified using the QIAquick PCR Purification Kit (28104, Qiagen) and used for library preparation with the AccelNGS 1S Plus DNA library kit (Swift Bioscience, 10096). Each of the library DNAs was sequenced using paired-end sequencing by Illumina NextSeq 500 platforms at the Columbia University Genome Center.

## ATAC-seq

Cells were collected, washed once with cold PBS, and lysed in cell lysis buffer (10 mM Tris-HCl pH 7.5, 10 mM NaCl, 3 mM MgCl$_2$, 0.1% NP-40, 0.1% Tween-20, 0.01% digitonin). After incubating on ice for 3 min, cells were washed once in wash buffer (10 mM Tris-HCl pH 7.5, 10 mM NaCl, 3 mM MgCl$_2$, 0.1% Tween-20), and centrifuge at 500× *g* for 10 min. Cell pellets were then incubated in transposition reaction buffer (10 mM Tris-HCl pH 7.6, 5 mM MgCl$_2$, 10% dimethylformamide, 0.1% Tween-20, 0.01% digitonin, 33% 1× PBS) with 1.5 μl pAG-Tn5 (15-1017, EpiCypher) at 37°C for 30 min. DNA was purified using the QIAquick PCR Purification Kit (28104, Qiagen). Library PCR was

performed using standard Illumina Nextera Dual Indexing primers. Libraries were sequenced using paired-end sequencing on Illumina NextSeq 500 platforms at the Columbia University Genome Center.

## CUT&RUN and ATAC-seq analysis

CUT&RUN and ATAC libraries were constructed and sequenced using paired-end method on Illumina platforms. Adaptor sequences of all raw reads were removed by Cutadapt (*Marcel, 2011*) and reads <10 nt were removed. CUT&RUN and ATAC-seq data were then mapped to mouse (mm10) reference genome by Bowtie2 (*Langmead and Salzberg, 2012*). Multi-mapped reads were removed using SAMtools (MAPQ <40) (*Li et al., 2009*) and duplicate reads were removed using Sambamba software (*Tarasov et al., 2015*). Read coverage in a bin of 1 bp was calculated from filtered bam files by deepTools2 (*Ramírez et al., 2016*) and then normalized with total filtered reads number into reads per million. Genome-wide correlation was performed by deepTools2 (*Ramírez et al., 2016*) in a bin of 5000 bp. Peaks were called by MACS (*Zhang et al., 2008*) by parameters 'macs2 callpeak -g mm -f BAMPE -p 1e-04 `--broad --broad-cutoff` 1e-04 `--llocal` 10000 –nolambda' and the cutoff of peak was p=0.0001.

Read density level surrounding gene promoters ([–3kb, 3kb] of TSS) or MCM2 peaks was calculated by featureCounts (*Liao et al., 2014*) and then normalized to reads per kilobase per million reads (RPKM). Heatmap clustering was performed by "ward.D2" (*Murtagh and Legendre, 2014*) method based on z score of log10(RPKM). To identify differential CUT&RUN peaks, peaks from both WT and mutant cells were first merged to a union pool and then read counts were calculated in the merged peaks by featureCounts (*Liao et al., 2014*). DESeq2 (*Love et al., 2014*) was then used to identify differential peaks by |log2 fold change|>1.

## RNA-seq analysis

Total RNA from WT and Mcm2-2A ESCs and NPCs were isolated from 1×106 cells using RNeasy Plus Mini kit (74136, Qiagen). RNA-seq libraries were prepared and deep sequenced in Columbia University Genome Center. Two replicates for each sample were sequenced. RNA-seq libraries were sequenced using paired-end method on Illumina platforms. The paired-end reads of WT and Asf1a KO ESCs and NPCs were downloaded from GSE114424. Adaptor sequences of all raw reads were removed by Cutadapt (*Marcel, 2011*) and reads <10 nt were removed. RNA-seq data were mapped to the mouse (mm10) reference genome by STAR software (*Dobin et al., 2013*). Gene expression levels were first calculated by featureCounts (*Liao et al., 2014*) to obtain read counts and then normalized with total mapped reads into RPKM. Differential expressed genes were identified by DESeq2 (*Love et al., 2014*) by using adjusted p-value <0.01 and |log2 fold change|>1. GO enrichment analysis was performed by 'cluserProfiler' (*Yu et al., 2012*) in a level 4 of 'biological process' functions.

## Statistical analyses

Data are presented as means ± SD. Differences between groups were evaluated using two-tailed unpaired Student's t test (noted in figure legends). Statistical analysis was performed in GraphPad Prism software (version 7). All tests were considered significant at p<0.05. For all the sequenced data analysis, the statistical test was performed using R software. Difference test between groups was evaluated by Wilcoxon signed-rank test. The center line is the medians of all data points, with the limits corresponding to the upper and the lower quartiles, respectively, and the whiskers representing the largest and smallest values. Where outliers were removed for plotting purposes, the removed data points were still used for statistical analyses.

## Acknowledgements

We thank Richard He for critical editing of the paper. DNA sequencing was performed in Columbia Genome Center with the support from the Herbert Irving Comprehensive Cancer Center at Columbia University and supported by NIH/NCI Cancer Center Support Grant P30CA013696. This work is supported by NIH grant GM R35118015 (to ZZ).

## Additional information

### Funding

| Funder | Grant reference number | Author |
|---|---|---|
| National Institute of General Medical Sciences | R35118015 | Zhiguo Zhang |

The funders had no role in study design, data collection and interpretation, or the decision to submit the work for publication.

### Author contributions

Xiaowei Xu, Conceptualization, Data curation, Formal analysis, Investigation, Methodology, Project administration, Resources, Writing – original draft, Writing – review and editing; Xu Hua, Conceptualization, Data curation, Formal analysis, Investigation, Project administration; Kyle Brown, Xiaojun Ren, Resources, Writing – review and editing; Zhiguo Zhang, Resources, Supervision, Funding acquisition, Formal analysis, Methodology, Writing – original draft, Project administration

### Author ORCIDs

Xiaowei Xu http://orcid.org/0000-0001-6050-9536
Xu Hua http://orcid.org/0000-0002-9775-4129
Xiaojun Ren http://orcid.org/0000-0002-3931-7625
Zhiguo Zhang http://orcid.org/0000-0002-9451-2685

### Decision letter and Author response

Decision letter https://doi.org/10.7554/eLife.80917.sa1
Author response https://doi.org/10.7554/eLife.80917.sa2

## Additional files

### Supplementary files

• MDAR checklist

### Data availability

Raw and processed sequencing data generated in the course of this study can be accessed via the GEO database with accession number: GSE203272.

The following dataset was generated:

| Author(s) | Year | Dataset title | Dataset URL | Database and Identifier |
|---|---|---|---|---|
| Xu X, Hua Xu, Brown Kyle, Ren X, Zhang Z | 2022 | Mcm2 promotes stem cell differentiation via its ability to bind H3-H4 | https://www.ncbi.nlm.nih.gov/geo/query/acc.cgi?acc=GSE203272 | NCBI Gene Expression Omnibus, GSE203272 |

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
