## [Editor Report]

This manuscript reports a novel role of Mcm2 licensing factor and helicase subunit of the Mcm2-Mcm7 complex in the differentiation of embryonic stem cells into neuronal lineages. A series of compelling experimental manipulations dissect the abnormalities in the formation of heterochromatin at pluripotent genes and the resolution of bivalent chromatin domains at lineage-specific genes in differentiation in response to mutation of the histone binding domain of Mcm2. These findings provide new insights into the replication-independent roles of Mcm2, and will be of interest to scientists working on development and embryonal cell differentiation.

---

## [Decision Letter]

**Decision letter after peer review:**

Thank you for submitting your article "Mcm2 promotes stem cell differentiation via its ability to bind H3-H4" for consideration by *eLife*. Your article has been reviewed by 2 peer reviewers, and the evaluation has been overseen by a Reviewing Editor and Jessica Tyler as the Senior Editor. The reviewers have opted to remain anonymous.

Essential revisions:

1. Further control data, analysis, and discussion on Mcm2 CUT&RUN experiments as suggested by both reviewers.

2. Clarification on the differences in Mcm2-2A mutant phenotype between ESCs and NPCs.

3. Potential redundancy between Mcm2 and histone chaperones.

4. Clarification on how Mcm2 may regulate deposition of newly synthesized or recycled histones.

5. Clarification on the role of reduced dormant origins reflected in the reduced density of Mcm2 at the origins in the differentiation to neuronal lineages.

6. The reviewers concur that investigating the mechanism of Mcm2 differential engagement with the bivalent chromatin domains in ESCs vs NPCs is an open question and may require extended experiments. It is recommended for authors to provide additional discussions on this issue if it is not feasible to be addressed experimentally in a revision.

*Reviewer #1 (Recommendations for the authors):*

As noted in the "Public Review" section, there is some concern about the low signal to noise of the Mcm2 CUT&RUN data that potentially impacts some of the conclusions. To resolve the concerns with these data, I suggest the following:

(1) Include negative control (IgG, etc) experiments in the figures and analyses. This may allow the authors to better determine which peaks are real.

(2) Attempt more stringent peak calling (e.g., SEACR). SEACR was designed specifically for CUT&RUN and can better deal with some of the issues (e.g., sparse background model) that can lead to artifacts in peak calling.

(3) Appropriately discuss/caveat these weaknesses in the conclusions regarding this data. To be clear, the authors' efforts to use a different antibody (FLAG) are admirable, and the similar results for Mcm2 and FLAG antibodies suggest the issue may be biological (i.e., Mcm2 binds broadly throughout chromatin and is not highly enriched at many loci). If that is the case, the other findings in the manuscript are still of considerable interest, but the Mcm2 binding data may need appropriate caveats.

Other recommendations:

(1) The scale bars for all heatmaps need better labels. For example, in Figure 3C, I assume the colors represent z-scores, but this is not clear. In Figure 4B, all of the heatmaps appear to share a color key, suggesting similar enrichment, but in other figures (such as Figure 5A), the scales are different for different proteins/histone modifications. Are these all expressed in reads per million or are different figures scaled differently?

(2) The authors should either go into greater detail to explain the surprising finding that Mcm2 is not enriched at origins but is enriched ~300 kb from origins, or these results should be omitted. As of now, the data are not well explained and somewhat distracting.

(3) The figure legend for Figure 6C contains a typo: it describes the ATAC-seq data to the left and CUT&RUN to the right. It is the opposite in the figure.

---

## [Author Response]

Essential revisions:1. Further control data, analysis, and discussion on Mcm2 CUT&RUN experiments as suggested by both reviewers.

We followed the reviewers’ suggestions and addressed their concerns related to this suggestion/requirement. Specifically, we performed two controls for Mcm2CUT&RUN, (1) CUT&RUN using antibodies against IgG, (2) CUT&RUN using Flag antibodies in wild type ES cells without Mcm2-Flag. We used the same parameters for peak calling on these two controls. As shown in (Figure 5—figure supplement 1B and 1C), we identified far few peaks in each of the two controls. Importantly, a large fraction of these peaks identified in the controls did not overlap with Mcm2 CUT&RUN peaks obtained using antibodies against Mcm2 or against the Flag epitope in cells expressing Flag-Mcm2. These results indicate that Mcm2 CUT&RUN peaks reflect Mcm2 binding. The relatively low signal to noise in Mcm2 CUT&RUN samples were likely due to dynamic nature of Mcm2 chromatin binding. Once loaded on chromatin during G1 phase of the cell cycle, it is known that the CMG helicase travels along DNA replication forks. Furthermore, Mcm2 CUT&RUN experiments were performed using asynchronous cells. These factors contribute to low signal to noise for Mcm2 CUT&RUN. We have also performed analysis using SEACR as suggested by the reviewers and we found that our original analysis is likely more reliable.

2. Clarification on the differences in Mcm2-2A mutant phenotype between ESCs and NPCs.

We discussed one potential reason for the differential impact of Mcm2-2A mutation on ES cells and in NPCs. It is known that excessive amounts of MCM2-7 helicase, greater than the number of replication origins, are loaded on chromatin during G1 phase of the cell cycle (Lei, Kawasaki et al. 1996, Edwards, Tutter et al. 2002). These excessive amount of MCM2-7 helicase, while not needed during unperturbed S phases of the cell cycle, helps fire dormant origins under replication stress (Woodward, Gohler et al. 2006). It is reported that ESCs contain more dormant origins than progenitor cells such as NPCs (Ge, Han et al. 2015). Consistent with this idea, we detected 13742 and 2686 Mcm2 CUT&RUN peaks, respectively, in wild type ES cells and NPCs. Furthermore, it is known that the depletion of Mcm5, whiling having little impact on stemness of ES cells, but dramatically affects differentiation (Ge, Han et al. 2015). However, I would like to point out that the differentiation defects detected in Mcm2-2A mutant cells are unlikely due to the impact of this mutant on dormant origins as Mcm2-2A mutant cells exhibited little defects in response to replication stress.

3. Potential redundancy between Mcm2 and histone chaperones.

Mcm2 is known to interact with Asf1 and this interaction is bridged by H3-H4. Furthermore, the Mcm2-Asf1 interaction is reduced in Mcm2-2A mutant cells. Therefore, we tested whether overexpression of Asf1a could rescue the differentiation defects of Mcm2-2A mutant cells. We found that Asf1a overexpression in Mcm2-2A cells did not rescue the differentiation defects of Mcm2-2A mutant cells (Figure 2—figure supplement 2B). However, overexpression of Mcm2 rescued the differentiation defects of Mcm2-2A cells.

To further investigate the relationship between Mcm2 and Asf1a, we knocked out Asf1a in both WT cells and Mcm2-2A mutant cells using CRISPR/Cas9 (Figure 2—figure supplement 2F and 2G) and analyzed the effects of Asf1a KO, Mcm2-2A and Mcm2-2A Asf1a KO double mutation on differentiation. We observed that Mcm2-2A Asf1a KO double mutant exhibited similar morphology as Mcm2-2A cells but not Asf1a KO cells after differentiation (Figure 2—figure supplement 2H). Moreover, both Mcm2-2A single and Mcm2-2A Asf1a KO double mutant cell lines showed defects in silencing of *Oct4*, while Asf1a KO cells did not show this defect (Figure 2—figure supplement 2I), implying Mcm2 has a role in silencing of Oct4, which is independent of its interaction with Asf1a. On the other hand, we observed that the defects in the induction of neural lineage gene expression (*Pax6* and *Sox21*) in Mcm2-2A cells and Asf1a KO cells as well as in Mcm2-2A cells Asf1a KO double mutant cells were quite similar (Figure 2—figure supplement 2I), indicating that Mcm2 functions in the same pathway with Asf1a for the induction of lineage specific genes upon differentiation. This idea was supported by analysis of transcriptomes changes in Asf1a KO and Mcm2-2A cells during differentiation (Figure 3—figure supplement 1). Based on these results and Mcm2 binding at bivalent chromatin domains, we propose that Mcm2 and Asf1a function together to resolve bivalent chromatin domain during exit from pluripotency. Furthermore, Mcm2 also has a role in silencing of pluripotent genes and this role is independent of Asf1a. We discussed three possible models for Mcm2 in silencing of pluripotent genes during differentiation (p23-24).

4. Clarification on how Mcm2 may regulate deposition of newly synthesized or recycled histones.

While it has been reported that Mcm2 may be involved in deposition of newly synthesized H3-H4 (Huang, Stromme et al. 2015), recent studies from my lab and Dr. Groth’s lab indicate that Mcm2-2A mutant cells are primarily defective in parental histone transfer in yeast and mouse ES cells (Gan, Serra-Cardona et al. 2018, Petryk, Dalby et al. 2018, Li, Hua et al. 2020). We discussed this point in the discussion in the revised manuscript.

5. Clarification on the role of reduced dormant origins reflected in the reduced density of Mcm2 at the origins in the differentiation to neuronal lineages.

We have now included this point in the discussion. Specifically, it is possible that defects in differentiation detected in Mcm2-2A mutant cells are likely linked to firing of dormant origins. It is known that excessive MCM2-7 complexes compared to active origins are loaded in G1 for activation of dormant origins, which are not needed for the S phase progression under normal conditions. However, dormant origins fire under replication stress. Furthermore, it has been shown that partial depletion of Mcm4 and Mcm5, two other subunits of MCM2-7 complex, while having little impacts on the self-renewal of ES cells, impairs the differentiation of ES cells including toward the neural lineage (Ge, Han et al. 2015). Therefore, it is possible that the differentiation defects of Mcm2-2A mutant arise from defects in dormant origins. Arguing against this idea, we have shown that Mcm2 and Asf1a functions together for the induction of lineage specific genes. Moreover, Mcm2-2A mutant cells are not sensitive to replication stress inducer, such as hydroxyurea. Furthermore, there is little evidence indicating that deletion of Asf1a affects dormant origin firing. We discussed these points in the discussion.

6. The reviewers concur that investigating the mechanism of Mcm2 differential engagement with the bivalent chromatin domains in ESCs vs NPCs is an open question and may require extended experiments. It is recommended for authors to provide additional discussions on this issue if it is not feasible to be addressed experimentally in a revision.

We thank the reviewers’ suggestion. We might have misled the reviewers that Mcm2 has different roles at bivalent chromatin domains of ES cells and NPCs in our original text. To clarify this situation, we have modified the discussion dramatically based on the new results indicating that Mcm2 and Asf1a function together for resolving bivalent chromatin domains during pluripotency exit. The role of Mcm2 in differentiation likely reflects its role in the differentiation process, but not necessarily due to the differential role of Mcm2 in ES cells and NPCs. We tried to point this out at the result as well as in the Discussion section in the revised manuscript.

Reviewer #1 (Recommendations for the authors):As noted in the "Public Review" section, there is some concern about the low signal to noise of the Mcm2 CUT&RUN data that potentially impacts some of the conclusions. To resolve the concerns with these data, I suggest the following:(1) Include negative control (IgG, etc) experiments in the figures and analyses. This may allow the authors to better determine which peaks are real.

We thank the reviewer’s suggestions. We have performed two controls for Mcm2 CUT&RUN, (1) CUT&RUN using antibodies against IgG, and (2) CUT&RUN using antibodies against the Flag epitope in wild type ES cells without Mcm2-Flag. We used the same parameters for peak calling on these two controls. As shown in (Figure 5—figure supplement 1B and 1C), we identified far few peaks in each of the two controls. Importantly, a large fraction of these peaks in the control samples did not overlap with Mcm2 CUT&RUN peaks using antibodies against Mcm2 and against the Flag epitope targeting Flag-Mcm2. These results indicate that Mcm2 CUT&RUN peaks, while showing low signal to noise ratio, likely reflect Mcm2 binding. The relatively low signal to noise of Mcm2 CUT&RUN peaks were likely due to dynamic nature of Mcm2 chromatin binding. Once loaded on chromatin during G1 phase of the cell cycle, it is known that the CMG helicase also travels along DNA replication forks. Furthermore, Mcm2 CUT&RUN experiments were performed using asynchronous ES cells because it is very challenging to synchronize ES cells. These factors may contribute to low signal to noise for Mcm2 CUT&RUN signals.

(2) Attempt more stringent peak calling (e.g., SEACR). SEACR was designed specifically for CUT&RUN and can better deal with some of the issues (e.g., sparse background model) that can lead to artifacts in peak calling.

We followed the reviewer’s suggestions and re-analyzed Mcm2 CUT&RUN datasets by using SEACR. We identified far more Mcm2 peaks using a "stringent" mode with its default numeric threshold of "0.01" (Author response image 1). Surprisingly, there are even more peaks in negative control samples (IgG/Flag CUT&RUN) than in Mcm2 CUT&RUN samples (Author response image 1 and 1C). However, when we examined several “peaks” in IgG CUT&RUN datasets identified by SEACR (0.01), we found that they were not real peaks (Author response image 1).

We then tried a more stringent cut off by using a threshold of “0.001”. While fewer peaks were identified than the threshold of "0.01" (Author response image 1), we still observed many false positive peaks recorded by SEACR (0.001) in IgG CUT&RUN sample (Author response image 1). Although most of these peaks in IgG samples were not overlapped with Mcm2 CUT&RUN peaks using the same method and cutoff (Author response image 1), we observed that several potentially real Mcm2 CUT&RUN peaks in Mcm2 CUT&RUN samples were not identified (Author response image 1, indicated by shadow).

Based on these results, we felt that the peak calling by MACS2 did not produce many false positive peaks in the negative control samples, and at the same time, the annotated peaks in Mcm2 CUT&RUN had high densities due to the stringent cutoff we set (*p*=0.0001). To address the reviewer’s concerns, we have included the negative control samples in revised figures and discussed the potential reasons for the relatively low signal to noise ratio for the Mcm2 CUT&RUN datasets (p16).

**Author response image 1. sa2fig1:** Comparison of peaks identified by two different programs, SEACR and MACS2. (A) A table displaying CUT&RUN peak numbers using either SEACR or MACS2 peak calling methods with numeric threshold indicated for each method. (B) The overlap of peaks between IgG CUT&RUN and Mcm2 CUT&RUN. SEACR was used for peak calling using a "stringent" mode with numeric threshold of "0.001". (C) Snapshots displaying IgG CUT&RUN and Mcm2 CUT&RUN density at two different loci in WT ES cells. Peaks identified using different methods/parameters were shown below each of the CUT&RUN track. Shadows indicate CUT&RUN peaks identified by SEACR (threshold=0.01) and MACS2 by not by SEACR (threshold=0.001).

(3) Appropriately discuss/caveat these weaknesses in the conclusions regarding this data. To be clear, the authors' efforts to use a different antibody (FLAG) are admirable, and the similar results for Mcm2 and FLAG antibodies suggest the issue may be biological (i.e., Mcm2 binds broadly throughout chromatin and is not highly enriched at many loci). If that is the case, the other findings in the manuscript are still of considerable interest, but the Mcm2 binding data may need appropriate caveats.

The reviewer’s point is well taken. We have now discussed the possible reasons for the low signal-to-noise ratio of Mcm2 CUT&RUN as described above (p16). Furthermore, we point out that cautions should be made for the interpretation of Mcm2 CUT&RUN results described below. (p16).

Other recommendations:(1) The scale bars for all heatmaps need better labels. For example, in Figure 3C, I assume the colors represent z-scores, but this is not clear. In Figure 4B, all of the heatmaps appear to share a color key, suggesting similar enrichment, but in other figures (such as Figure 5A), the scales are different for different proteins/histone modifications. Are these all expressed in reads per million or are different figures scaled differently?

Thank the reviewer for suggestions. We have now included all the definition of the heatmap scale bars in respect figure legends. To summarize, in Figure 3C, Figure 4—figure supplement 1C and 1D and Figure 6—figure supplement 2A, color scales represent z-scores. In Figure 4B, Figure 5A, Figure 5—figure supplement 1D, Figure 6C and 6E and Figure 6—figure supplement 2B, 2E and 2G, the scale represents reads per million.

(2) The authors should either go into greater detail to explain the surprising finding that Mcm2 is not enriched at origins but is enriched ~300 kb from origins, or these results should be omitted. As of now, the data are not well explained and somewhat distracting.

We thank the reviewer for suggestions. We agreed with the reviewer that the results, while interesting, are distracting. We followed the reviewer’s suggestion and removed the results in the revised manuscript.

(3) The figure legend for Figure 6C contains a typo: it describes the ATAC-seq data to the left and CUT&RUN to the right. It is the opposite in the figure.

We have now corrected the typo.

References:

Edwards, M. C., A. V. Tutter, C. Cvetic, C. H. Gilbert, T. A. Prokhorova and J. C. Walter (2002). "MCM2-7 complexes bind chromatin in a distributed pattern surrounding the origin recognition complex in *Xenopus* egg extracts." J Biol Chem 277(36): 33049-33057.

Foltman, M., C. Evrin, G. De Piccoli, R. C. Jones, R. D. Edmondson, Y. Katou, R. Nakato, K. Shirahige and K. Labib (2013). "Eukaryotic replisome components cooperate to process histones during chromosome replication." Cell Rep 3(3): 892-904.

Gan, H., A. Serra-Cardona, X. Hua, H. Zhou, K. Labib, C. Yu and Z. Zhang (2018). "The Mcm2-Ctf4-Polalpha Axis Facilitates Parental Histone H3-H4 Transfer to Lagging Strands." Mol Cell 72(1): 140-151 e143.

Ge, X. Q., J. Han, E. C. Cheng, S. Yamaguchi, N. Shima, J. L. Thomas and H. Lin (2015). "Embryonic Stem Cells License a High Level of Dormant Origins to Protect the Genome against Replication Stress." Stem Cell Reports 5(2): 185-194.

Huang, H. D., C. B. Stromme, G. Saredi, M. Hodl, A. Strandsby, C. Gonzalez-Aguilera, S. Chen, A. Groth and D. J. Patel (2015). "A unique binding mode enables MCM2 to chaperone histones H3-H4 at replication forks." Nature Structural & Molecular Biology 22(8): 618-626.

Lei, M., Y. Kawasaki and B. K. Tye (1996). "Physical interactions among Mcm proteins and effects of Mcm dosage on DNA replication in *Saccharomyces cerevisiae*." Molecular and Cellular Biology 16(9): 5081-5090.

Li, Z., X. Hua, A. Serra-Cardona, X. Xu, S. Gan, H. Zhou, W. S. Yang, C. L. Chen, R. M. Xu and Z. Zhang (2020). "DNA polymerase α interacts with H3-H4 and facilitates the transfer of parental histones to lagging strands." Sci Adv 6(35): eabb5820.

Petryk, N., M. Dalby, A. Wenger, C. B. Stromme, A. Strandsby, R. Andersson and A. Groth (2018). "MCM2 promotes symmetric inheritance of modified histones during DNA replication." Science 361(6409): 1389-1391.

Woodward, A. M., T. Gohler, M. G. Luciani, M. Oehlmann, X. Q. Ge, A. Gartner, D. A. Jackson and J. J. Blow (2006). "Excess Mcm2-7 license dormant origins of replication that can be used under conditions of replicative stress." Journal of Cell Biology 173(5): 673-683.